SPECIAL ISSUE
**THE EXTRACELLULAR ENVIRONMENT**

# An essential role for actinotrichia in zebrafish fin patterning and courtship behavior

Paulina Hanzelova, Connor Baird, Bidemi Keshinro, Reeham Kadhom*, Robert L. Lalonde‡ and Marie-Andrée Akimenko§

## ABSTRACT

A key difference between tetrapod limb buds and teleost fin buds is the presence of rigid actinotrichia fibers that guide the migrating cells contributing to ray formation. Major structural components of actinotrichia are encoded by fish-specific actinodin (And) genes, which were lost in tetrapods. To investigate the consequences of this loss during the fin-to-limb transition, we generated deletions in zebrafish *and1* and *and2* using CRISPR/Cas9 mutagenesis. Double mutants ($and1^{-/-}and2^{-/-}$) lack actinotrichia. Embryos and larvae have reduced fin fold size, with disorganized cell migration. In adults, all fin fold-derived skeletal structures are disrupted, including the rays of all fins, as well as the caudal fin endoskeleton. Surprisingly, double mutant males fail to breed, despite being fertile. Video analysis revealed that defects in the fins of males impair their ability to stimulate egg release. Our findings highlight the role of actinotrichia in both fin patterning and zebrafish courtship. We propose that actinodin gene maintenance is under strong selection in fish with similar courtship. We speculate that the loss of actinodin genes and a shift in courtship strategy may have coincided during tetrapod evolution.

KEY WORDS: Actinotrichia, Fin development, Fin-to-limb transition, Zebrafish courtship, Vertebrate evolution

## INTRODUCTION

The emergence of vertebrates from water onto land involved the evolution of limbs suited for terrestrial life from ancestral paired fins. This fin-to-limb transition involved the elaboration of the appendage endoskeleton, and the reduction and loss of the dermoskeletal fin rays (Long and Gordon, 2004; Shubin et al., 2004, 2006, 2014). Tetrapod limbs are developmentally homologous to the paired fins of fish, with their initial development being strikingly similar; both structures emerge as buds from the lateral plate mesoderm (Mercader, 2007). However, one essential difference is the fate of the apical ectodermal ridge (AER), a specialized epithelial layer at the distal margin of

Department of Biology, University of Ottawa, Ottawa, Ontario K1N 6N5, Canada. *Present address: Dalhousie Medical School, Halifax, Nova Scotia B3H 4R2, Canada. ‡Present address: Yale School of Medicine, Yale University, New Haven, CT 06510, USA.

§Author for correspondence (marie-andree.akimenko@uottawa.ca)

P.H., 0009-0005-9339-8066; M.-A.A., 0000-0002-1905-7731

the bud (Saunders, 1948; Summerbell, 1974). In both paired fin and limb buds, the AER maintains mesenchymal cells in an undifferentiated and proliferative state (Grandel and Schulte-Merker, 1998; Grandel et al., 2000; Kawakami et al., 2003). However, after skeletal progenitors are specified in the limb bud, the AER regresses (Saunders, 1948; Pizette and Niswander, 1999), whereas in teleost fin buds it persists and forms a fin fold (FF), giving rise to the external parts of the fin dermoskeleton (Grandel and Schulte-Merker, 1998; Yano et al., 2012). The paired pectoral and pelvic fin dermoskeletons arise from the pectoral and pelvic FFs, respectively, whereas the dorsal, anal and caudal fin dermoskeletons arise from the median FF, which is situated medially along the posterior half of the body. Along the proximodistal axis of each FF lie two parallel arrays of rigid actinotrichia fibers (Fig. S1A) that provide structural support and a scaffold for distally migrating mesenchymal cells (Wood, 1982; Wood and Thorogood, 1984; Dane and Tucker, 1985), which include osteoblast precursors that will form the bony rays (Nakamura et al., 2016; Lee et al., 2013). Each ray consists of two subepidermal concave hemirays composed of bone segments regularly spaced by fibrous joints (Fig. S1B) (Lee et al., 2013; Becerra et al., 1983; Marí-Beffa et al., 1989). As the rays develop, actinotrichia fibers are continually synthesized distally and degraded proximally by osteoclasts, resulting in the maintenance of their location at the tips of the rays (Kuroda et al., 2024; Marí-Beffa et al., 1989; Durán et al., 2011; Géraudie and Landis, 1982). In each mature ray, the actinotrichia form two arrays on the internal surface of the distalmost bone segments (Fig. S1B).

Actinotrichia were first described as unmineralized fibers composed of elastoidin, a complex containing collagen and a tyrosine-rich non-collagenous compound (Krukenberg, 1885; Gross and Dumsha, 1958). Elastoidin is distinguishable from collagen by its transparency (Ellis and McGavin, 1970) and tendency to form long needle-like fibers that behave as one single unit (McGavin and Pyper, 1964). Actinotrichia are huge fibers relative to the tissue they occupy, reaching up to 100 μm in larval zebrafish FFs (Kuroda et al., 2018) and increasing in length during development to 200-300 μm at the tips of adult zebrafish caudal rays (Nakagawa et al., 2022). Although their size varies depending on developmental stage (Durán et al., 2011; Nakagawa et al., 2022; Kuroda et al., 2018), actinotrichia have a characteristic distribution in developing FFs and in the adult rays. Before ray formation, actinotrichia form two arrays of fibers, radially arranged approximately parallel to one another, spanning the entire proximodistal length of the FF (van den Boogart et al., 2012). In adults, they overlap with the one or two distalmost bone segments of the rays, and extend past the bone, arranged in a distinct fan shape (Durán et al., 2011; Pfefferli and Jaźwińska, 2015). These distributions give a texture to the overlaying epidermis that can be visualized with light microscopy (Fig. S1). A candidate for the non-collagenous compound of actinotrichia was identified in zebrafish as actinodin (And), encoded

by the Actinodin gene family (Zhang et al., 2010). In teleosts, there are four paralogous And genes (*and1*, *and2*, *and3* and *and4*) (Padhi et al., 2004; Zhang et al., 2010).

And genes have been identified in all gnathostomes studied thus far, except in tetrapods (Zhang et al., 2010; Biscotti et al., 2016; Amemiya et al., 2013). The loss of the And genes may be related to the vertebrate fin-to-limb transition during the late Devonian Period (Zhang et al., 2010). Morpholino-mediated gene knockdown of both *and1* and *and2* results in the temporary loss of actinotrichia in zebrafish embryos, leading to defects in FF morphology and in cell migration due to the loss of the scaffold that actinotrichia usually provide (Zhang et al., 2010). Single-knockdown morphants retain actinotrichia and have normal FF morphology, likely due to genetic compensation between paralogs. Owing to the temporary nature of morpholino-mediated gene knockdown, it was not possible to observe the consequences of actinotrichia loss on later stages of fin development. We therefore generated deletion mutations in *and1* and *and2* using CRISPR/Cas9 genome editing to study the role of actinotrichia in fin development beyond embryonic stages. Like the *and1/and2* co-morphants, the double mutants lack actinotrichia and have consequent FF defects. These defects accumulate to the malformation of the structures forming within the FF, including the bony rays of all fins and the caudal fin endoskeleton. Interestingly, we show that double mutant males are unable to breed because the defects in their pectoral fins impair proper courtship behavior. We propose that the maintenance of actinotrichia is under strong selection in fish that use their pectoral fins in courtship.

## RESULTS

### Deletions in *and1* and *and2* cause loss of actinotrichia, leading to embryonic fin fold defects

CRISPR-Cas9 genome editing was used to induce a 669 bp deletion in *and1* and a 269 bp deletion in *and2* (Fig. 1A; Fig. S2). The partial deletion in *and1* results in the loss of exon 2, which contains the translation start codon. The partial deletion in *and2* leads to a truncation of the protein and removes 8 out of 10 repeats from the C-terminus of the protein; these repeats are thought to be important for actinotrichia formation (Zhang et al., 2010). At 2 days post fertilization (dpf), *and1* and *and2* are both strongly expressed in the developing pectoral and median FF of wild-type embryos, shown by *in situ* hybridization of *and1* and *and2* antisense riboprobes (Zhang et al., 2010) (Fig. 1B). The single homozygous mutants $and1^{-/-}and2^{+/+}$ and $and1^{+/+}and2^{-/-}$ showed no expression in the FFs for the deleted gene (Fig. 1B), likely due to nonsense-mediated mRNA decay (Wittkopp et al., 2009). The double homozygous ($and1^{-/-}and2^{-/-}$) mutant embryos (referred to hereafter as double mutants) had no expression in the FFs of either *and1* or *and2* (Fig. 1B) after 20 min of staining, when they are normally both strongly expressed in the FFs. Trace *and2* expression in the median FF of *and2* single mutants and double mutants was detected after prolonged staining (Fig. S3A); however, this is not unexpected, given the *and2* deletion results in a truncated protein, but may still encode a transcript (Fig. 1A). RT-qPCR analysis of relative *and1* and *and2* expression levels showed significant decreases in the double mutants relative to wild-type siblings at 5 dpf (Fig. 1C). There was no significant upregulation in *and3*. Interestingly, *and4* was significantly upregulated (Fig. 1C); however, the expression level is low for both wild-type and double mutant larvae (Fig. S3B). Wild-type sibling and single mutant FFs had actinotrichia, whereas double mutants did not (Fig. 2A). Double mutant pectoral and median FFs had morphological defects, shown by an irregular

margin and wrinkling of the tissue (Fig. 2A). And1 and type II collagen (Col2) immunostaining at 5 dpf showed that signals localized to actinotrichia fibers in wild-type FFs (Fig. 2B). And1 was not detected in double mutant larvae, and Col2 aggregated instead of incorporating into actinotrichia (Fig. 2B; Fig. S3C). Morphometric analysis until 7 dpf revealed that the overall size of the FFs is reduced in double mutants relative to wild-type siblings (Fig. S4B,C), suggesting that the lack of actinotrichia in the FFs disrupts their distal outgrowth during development. Size reduction was also restricted to the FFs, as overall body size and endoskeletal disc (ED) size were unaffected in double mutant embryos and larvae (Fig. S4A,D,E). FF cell migration was evaluated using the ET37 enhancer trap line, expressing GFP in the migrating FF mesenchyme, in the double mutant genetic background (Zhang et al., 2010; Choo et al., 2006). As mesenchymal cells migrate along the actinotrichia in rows in the wild-type FFs, they take on an elongated shape (Zhang et al., 2010) (Fig. 2C). In the double mutant FFs, the cells did not follow this linear migration pattern and were arranged in a disorganized web (Fig. 2C). This is consistent with actinotrichia providing a structural scaffold for migrating mesenchymal cells. The defects observed in the double mutant embryos support previous observations of the *and1/and2* co-morphants (Zhang et al., 2010), showing that the loss of function of both *and1* and *and2* leads to the loss of actinotrichia in the FFs, and has consequences on FF morphology and development.

### Loss of actinotrichia leads to defects in adult fins

The stable mutant lines allow the observation of the long-term effect of *and1* and *and2* loss of function, and its impact on the adult fins. Adult single mutants had no observable fin defects (Fig. S5A). All fins were affected in the double mutant adults, as seen on live fin images (Fig. 3A) and by Alizarin Red staining of the fin dermoskeleton (Fig. 3B). Double mutants had shorter and wider rays with irregularly spaced bone segments and an overall wavy appearance (Fig. 3B). Lengths and widths of all fins normalized to standard length (SL) were found to be significantly reduced in double mutants relative to wild-type siblings (Fig. 3C,D). The number of rays in each fin was also significantly reduced in double mutants (Fig. 3E). Whereas soft rays normally bifurcate into two sister rays, the rays of the double mutant fins rarely bifurcated (pectoral and pelvic fins) or did not bifurcate at all (dorsal, anal and caudal fins) (Fig. 3F). Fin ray length, number and level of bifurcation influence the overall geometry of the fin, which may explain the reduced size and less fan-like appearance of the double mutant fins. The effect was most dramatic in the caudal fin, which normally presents with a distinct forked geometry (Desvignes et al., 2022) due to longer peripheral rays making up the dorsal and ventral lobes of the fin, and relatively shorter central rays (Fig. S5B).

Normally, soft rays are straight and distally tapering, with evenly spaced joints connecting each bone segment (Fig. S1B; Fig. 4A). The double mutant rays were deformed, suggesting defects in bone patterning (Fig. 4B), and were wider and wavy, with irregularly spaced joints connecting each bone segment (Fig. 4B). At the tips of wild-type rays, a fan of actinotrichia supports the distal epidermal tissue of the fin (Fig. 4C,E). The double mutant rays end bluntly, rather than tapering distally to the extremity of the fin (Fig. 4D,F). The distalmost bone segments of double mutant rays are also cracked (Fig. 4F). The inter-ray tissue receded along the margin of the fin (Fig. 4D,F). And1 (Fig. 4G) and Col2 (Fig. 4I) immunostaining of the tips of the wild-type rays showed signal localized to the actinotrichia fibers, while, in double mutants, no And1 signal is visible (Fig. 4H), while the Col2 signal faintly

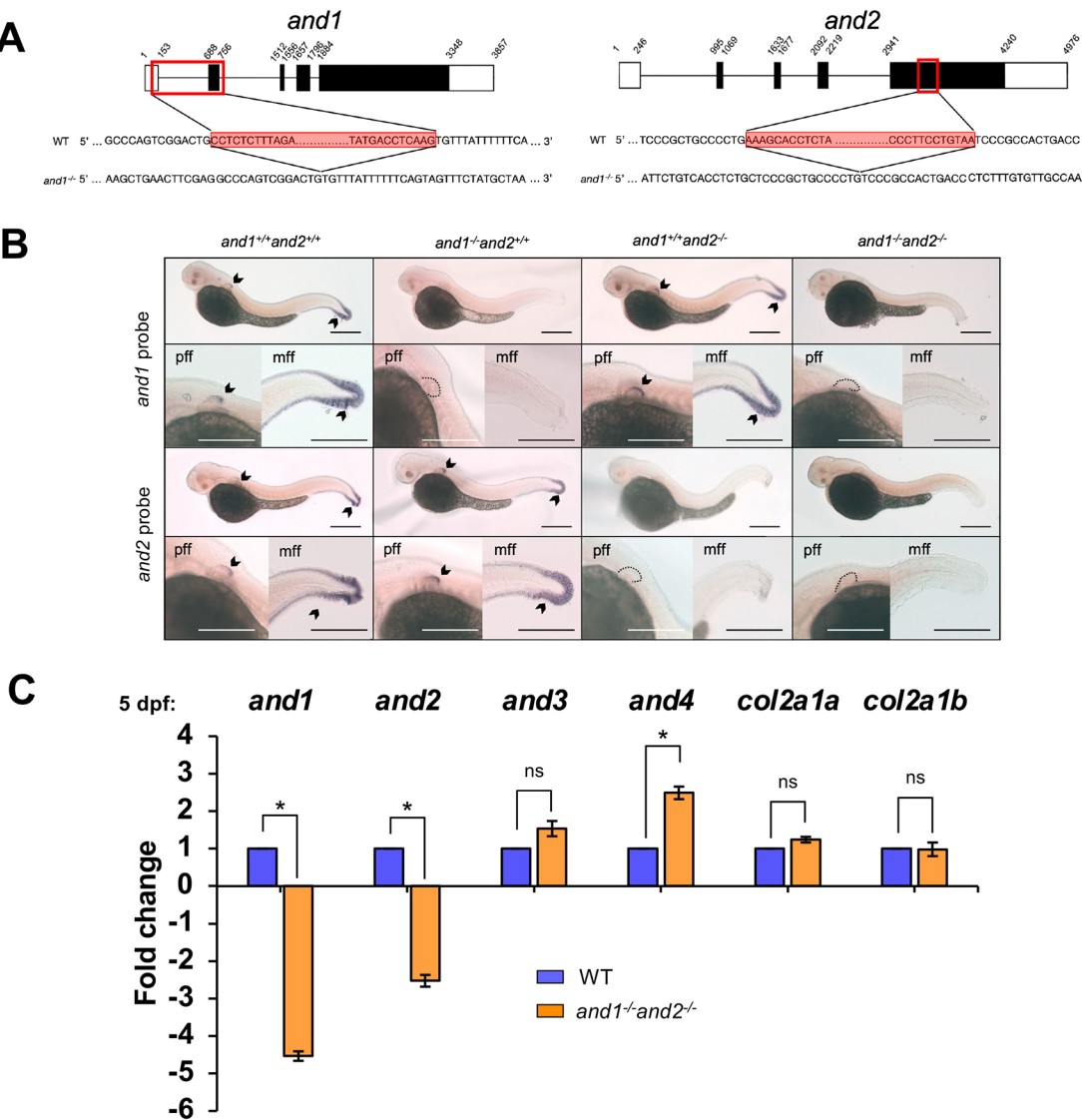

**Fig. 1. Deletions in *and1* and *and2* lead to absent or reduced expression.** (A) Schematic representation of deletions (red outlines) in *and1* and *and2*. Black rectangles indicate the coding regions; white rectangles represent the untranslated exons of the genes. Numbers refer to base pairs in the gene sequence. The region around the deletion sites (red highlight) in wild-type and mutated sequences is shown below. (B) Whole-mount *in situ* hybridization of *and1* and *and2* antisense riboprobes of wild-type ($n_{and1}$=10; $n_{and2}$=10), $and1^{-/-}and2^{+/+}$ ($n_{and1}$=4; $n_{and2}$=5), $and1^{+/+}and2^{-/-}$ ($n_{and1}$=6; $n_{and2}$=4) and $and1^{-/-}and2^{-/-}$ ($n_{and1}$=5; $n_{and2}$=5) embryos at 2 dpf. Embryos were stained for 20 min. Black arrowheads indicate purple staining in median fin folds (mff) and pectoral fin folds (pff). Pectoral fin folds lacking expression are emphasized with a black dotted outline. Scale bars: 250 μm. (C) Relative fold change of *and1*, *and2*, *and3*, *and4*, *col2a1a* and *col2a1b* expression in wild-type sibling and double mutant tails at 5 dpf (*n*=30 tails per biological replicate). Wild-type expression level was set to 1. Fold change was determined by RT-qPCR using the ΔΔCq method. Error bars indicate standard error of ΔΔCq. *$P$<0.05 (unpaired two-tailed *t*-test).

accumulates at the tip of the ray in the absence of actinotrichia (Fig. 4J; Fig. S5C). In the wild type, the actinotrichia were arranged in two arrays, overlapping with the distalmost bone segments and extending past the bone (Figs S1B and S6A,B,D). There were no actinotrichia detected between the double mutant hemirays or distally beyond them (Fig. S6C,E). The double mutant hemirays were wider (Figs 4B and S6G) and each hemiray was thicker than in the wild type (Fig. S6F,G).

Altogether, these analyses show that, while deletions within only one of the two And genes of interest do not result in obvious fin defects, deletions in both *and1* and *and2* result in their loss of function, resulting in a lack of actinotrichia. This shows that the And genes are necessary for actinotrichia synthesis, which in turn is important for proper ray patterning.

## Actinotrichia influence the development of all skeletal structures that form within the fin fold

Unlike the bones of the dermoskeletal rays, which form through intramembranous ossification, the bones of the fin endoskeleton develop through endochondral ossification, via a cartilage intermediate (Hall, 2005). In the caudal fin, one parhypural (Ph) and five hypurals (H1-H5) branch from the posterior vertebrae and articulate distally with the fin rays (Bird and Mabee, 2003).

In addition to the defects in the fin dermoskeletons, double mutants also had variable fusions of the parhypural and hypural bones (referred to collectively as hypural elements) in the caudal fin, revealed by Alizarin Red staining in adults (Fig. S7A1,B1). Although natural non-pathological variation has been reported in the teleost caudal fin endoskeleton (Koumoundouros et al., 2001;

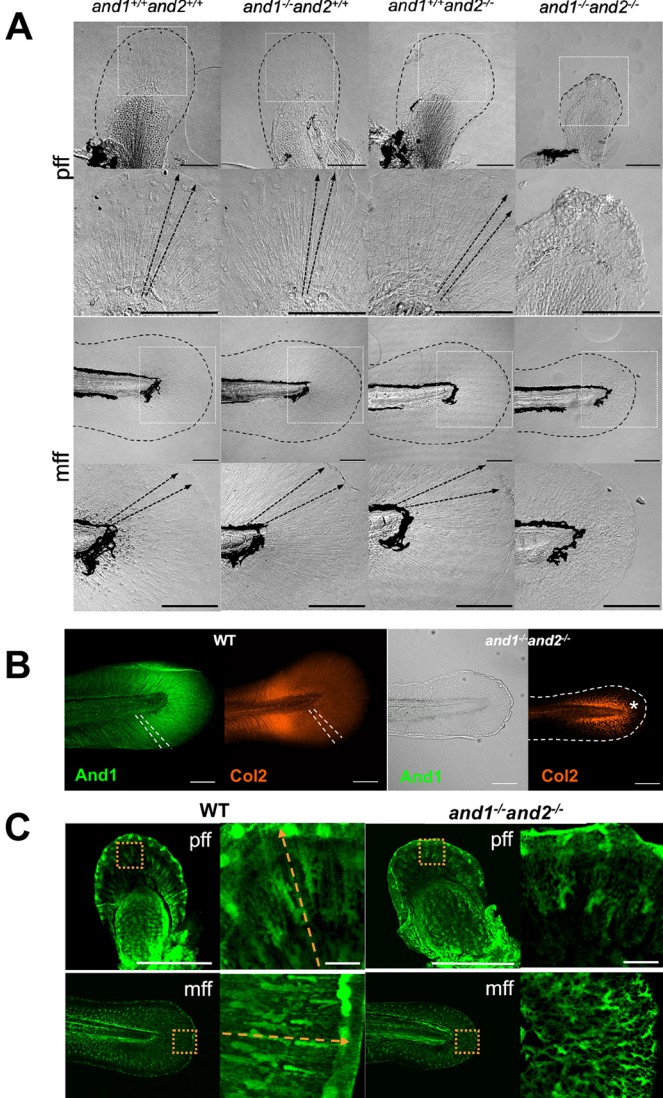

**Fig. 2. Simultaneous deletion of _and1_ and _and2_ leads to absence of actinotrichia and fin-fold defects.** (A) Transmitted light differential interference (TD) images of pectoral (pff) and median (mff) fin folds of wild type (_n_=5 larvae), _and1⁻/⁻and2⁺/⁺_ (_n_=3 larvae), _and1⁻/⁻and2⁺/⁺_ (_n_=3 larvae) and double mutants (_n_=5 larvae) at 5 dpf. Area outlined by solid white line is shown at a higher magnification below. The dashed black line outlines the fin fold (FF). Actinotrichia (black dotted arrows) are present in wild type but absent in double mutants. Wrinkling (white asterisk) is visible in the double mutant FF tissue. Scale bars: 100 μm. (B) And1- and Col2-immunostained wild-type sibling (_n_=6 larvae) and double mutant (_n_=6 larvae) posterior median FFs at 5 dpf. Dashed lines represent actinotrichia in the wild-type siblings. Actinotrichia are absent in the double mutant mff, shown by lack of And1 signal (a merged image of the fluorescent and transmitted light channels is shown), and by disorganized and aggregating Col2 (white asterisk). The FF margin is outlined in the Col2-immunostained double mutant. Scale bars: 100 μm. (C) Maximum intensity projection confocal images of ET37 wild-type (_n_=5 larvae) and double mutant (_n_=4 larvae) pff and mff at 5 dpf. Area outlined is shown at higher magnification on the right. Orange dashed arrows indicate the orientation of actinotrichia in the wild-type siblings. Scale bars: 200 μm; 50 μm in magnified image.

Bensimon-Brito et al., 2012; Witten and Hall, 2015; McDowall, 1999), double mutants had significantly fewer numbers of hypural elements than wild-type siblings due to fusions between them. Wild-type siblings had an average of 5.9±0.125 elements, whereas

double mutants had 4.5±0.189 elements (Fig. S7C). Double mutants had dorsal and ventral connective tissue plates (Desvignes et al., 2018, 2022) at the distal ends of their hypural elements (Fig. S8B1), like wild-type siblings (Fig. S7A1). Fused elements seemingly occupy the same space that they would if they were unfused (Fig. S7A1,B1), suggesting that this decrease in number is not due to missing hypurals. Hypural fusion is further supported by analysis of their development. Although _and1_ and _and2_ are not known to contribute to the development of the hypurals (Fig. S8A), Col2 contributes to the development of both actinotrichia and the hypural cartilage templates (Durán et al., 2011, 2015; Dale and Topczewski, 2011). Col2 immunostaining shows that hypurals were forming within the FF, initiating near the base of the actinotrichia (Fig. 5A) and growing between the two actinotrichia arrays (Fig. 5B,I,J; Fig. S8B). The orientation of cartilage growth seemed to follow the orientation of the fibers. We therefore examined the impact of the absence of actinotrichia on hypural development. Double mutants had normal initiation of hypural development, suggesting that actinotrichia are not influencing the positioning of initial mesenchymal condensation (Fig. 5C). It was during the growth of the cartilage template that disorganization and fusions appeared, prior to ossification, with no apparent developmental delay (Fig. 5D). Col2 aggregates were localized at the caudal tip of the notochord (Fig. 5C,D), away from the developing hypurals (Fig. S8C). Additionally, while double mutants and wild-type siblings showed a similar initial distribution of melanocytes around the posterior part of the trunk at 4.0 mm SL (Fig. 5E,G), the pattern later became disorganized in the double mutant FF region (Fig. 5F,H).

In contrast to the caudal fin endoskeleton, which develops within the actinotrichia-populated median FF, the dorsal and anal fin endoskeletons develop in the trunk, and the pectoral and pelvic fin endoskeletons develop within the fin bud mesenchyme, with the pectoral fin endoskeleton developing in a specialized endoskeletal disc (Bird and Mabee, 2003; Grandel and Schulte-Merker, 1998). The caudal fin endoskeleton is, therefore, the only fin endoskeleton that develops in the presence of actinotrichia; the other fin endoskeletons develop proximal to the domain of the fibers and, as such, were not perturbed in the double mutants (Fig. S7D-G). The reduced numbers of fin rays in the double mutants articulated with normal numbers of endoskeletal elements, resulting in an abnormal dermoskeletal/endoskeletal interface (Fig. S7B2,B3).

The hypural fusions seemingly result from a malformed cartilage template, likely influenced by the lack of the structural scaffold normally provided by actinotrichia. Therefore, the actinotrichia influence the development of all the FF-derived skeletal structures, which, in the case of the caudal fin, includes the endoskeleton.

### Absence of actinotrichia impairs male reproductive ability

While the known roles of actinotrichia are related to fin development, we noticed that the double mutant males are unable to breed. Whereas double mutant females bred when crossed with males of other genotypes, double mutant males yielded no embryos, even when crossed with wild-type females (Fig. S9A). Furthermore, females became eggbound when housed with double mutant males. Double mutants had no obvious morphological defects outside their fins (Figs S4A and S9B,C), e.g. the urogenital region appeared morphologically normal (Fig. S9C). Double mutant male fertility was assessed using _in vitro_ fertilization (IVF) of wild-type eggs with sperm from sexually mature wild-type and double mutant males. Regardless of fertilization efficiency, which depends on many factors during IVF (Hagedorn and Carter, 2011), we found that double mutant male sperm could fertilize wild-type eggs (Fig. 6A); therefore,

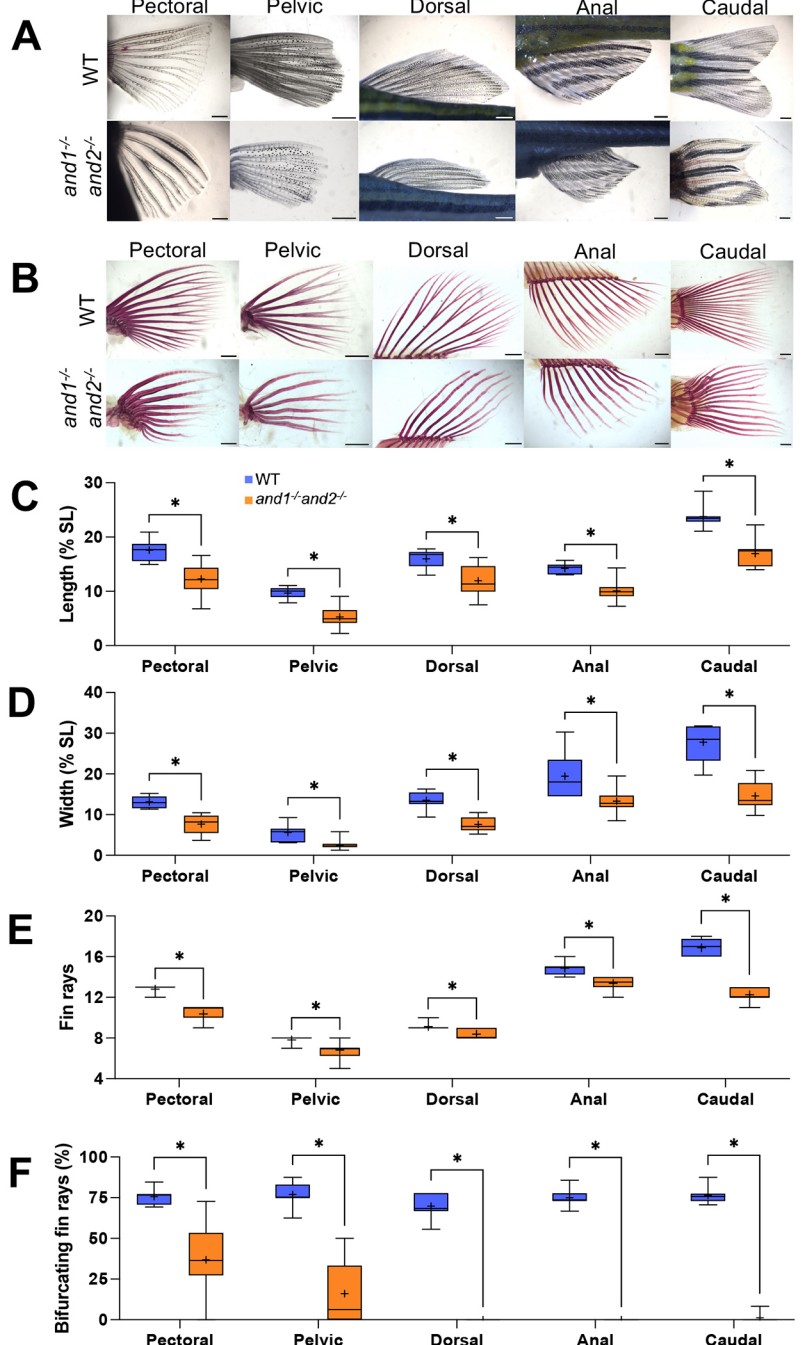

**Fig. 3. The lack of actinotrichia leads to morphological defects in the adult fin.** (A,B) Bright-field images of live (A) and Alizarin Red-stained (B) adult wild-type sibling (*n*=8 fish) and double mutant fins (*n*=8 fish) at 10 months post-fertilization ranging between 28 mm and 30 mm standard length (SL). Double mutant dorsal and caudal fins are also shown in Fig. S7B1,B2. Scale bars: 1 mm. (C,D) Mean (+) lengths (C) and widths (D) of fins in wild-type siblings (*n*=7 fish) and double mutants (*n*=12 fish) at 140 dpf as a percentage of SL, ranging between 29 and 33 mm SL. Length and width were measured at the longest and widest points of each fin, respectively. (E,F) Number of rays (E) and percentage of bifurcating rays (F) in each fin in wild-type siblings (*n*=8 fish) and double mutants (*n*=8 fish) at 10 months post-fertilization ranging between 28 and 30 mm SL. \**P*<0.05 (unpaired two-tailed Mann–Whitney U-test, FDR=1%). Box plots show means (+), first and third quartiles (box), and lowest and highest values (whiskers).

the inability of the double mutants to breed is not due to gamete deficiency. Successful reproduction in many fish depends on the execution of a series of distinct, predictable courtship behaviors (Kime, 1998; Darrow and Harris, 2004; Zempo et al., 2021). We therefore compared the courtship behavior of wild-type and double mutant males paired with wild-type females by video analysis, categorizing behaviors according to Zempo et al., (2021). In laboratory settings, zebrafish courtship is initiated by chasing of the female by the male (Chasing). The male escorts the receptive female to their preferred spawning site in shallow water (Escorting) (Darrow and Harris, 2004) and they encircle each other (Encircling) (Zempo et al., 2021). The Chasing-Escorting-Encircling may repeat for several minutes. Eventually, the male quivers his body near the female (Quivering) (Zempo et al., 2021). Finally, the male

physically squeezes the female using his body (Wrap-around) to release her eggs (Spawning). The Wrap-around behavior includes three additional events: in response to Quivering, the female stops swimming and bends her body adjacent to the male (Freezing); the male places a pectoral fin underneath the female (assumed to stimulate egg release) (Zempo et al., 2021; Kang et al., 2013) and hooks his trunk around her (Hooking); the female moves forward from within the grasp of the male, applying pressure to her belly (Squeezing). Egg release after this step is evident, and sperm release is assumed to occur simultaneously (Zempo et al., 2021). The role of the male pectoral fins in courtship may explain the inability of the double mutant males to breed. We therefore compared the courtship behavior of wild-type and double mutant males paired with wild-type females by video analysis. Wild-type sibling males paired

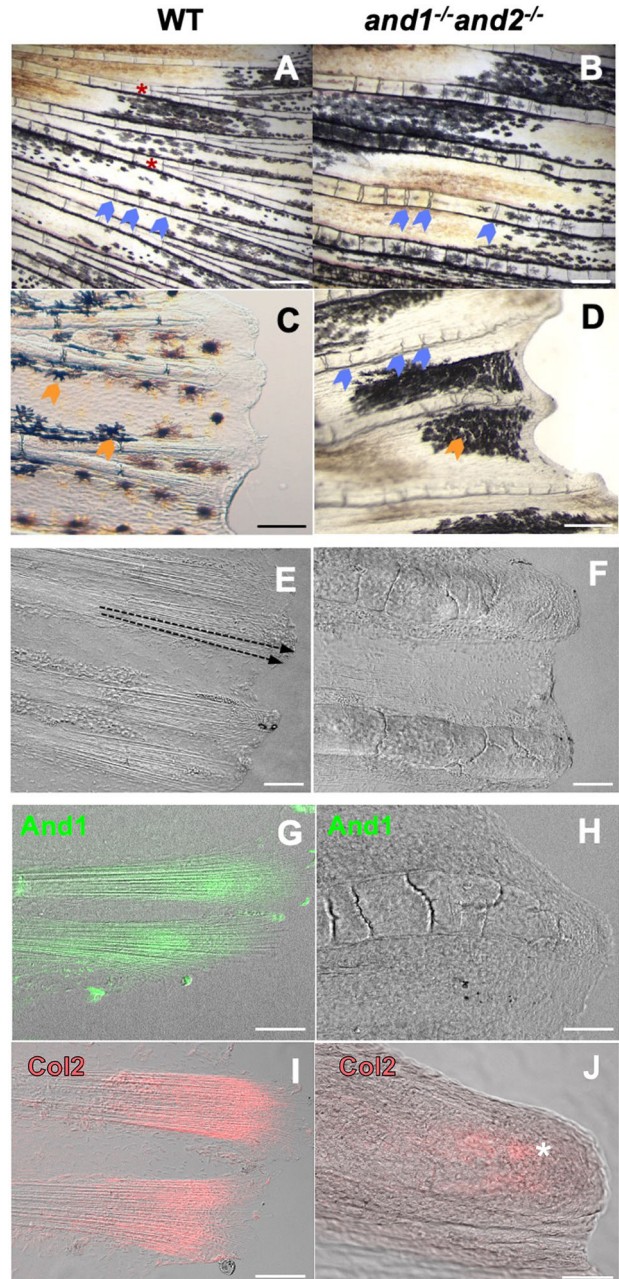

**Fig. 4. Loss of actinotrichia impairs ray patterning.** (A-D) Bright-field images of wild-type (*n*=10 fins) (A,C) and double mutant (*n*=10 fins) (B,D) caudal fin rays. Joint spacing (blue arrowheads) is regular in wild-type rays (A), but irregular in double mutant rays (B). Bifurcations (red asterisks) are present in wild-type rays (A,C) but absent in double mutant rays (B,D). The wild-type rays taper (C), while, in the double mutant, rays end bluntly, with the inter-ray region receding (D). Melanocytes show a normal distribution in wild-type fins (orange arrowheads), but seem aggregated in the double mutant. Scale bars: 250 µm. (E,F) Transmitted light differential interference (TD) images of wild-type (E) (*n*=10 fins) and double mutant (F) (*n*=10 fins) caudal fin ray tips. Dotted arrows represent the direction of the actinotrichia located at the tip of each ray (E). No visible actinotrichia are found in the double mutant rays (F). The tips of double mutant rays end bluntly (F), rather than tapering distally (E). Scale bars: 50 µm. (G-J) Merged confocal and TD images of And1- (G,H) and Col2-immunostained (I,J) wild-type sibling (G,I) and double mutant (H,J) caudal fin ray tips. And1 signal is localized to actinotrichia fibers at the tips of wild-type rays (G) (*n*=5 fins), but is absent in double mutant rays (*n*=5 fins). Col2 signal also localizes to the actinotrichia in wild-type rays (I) (*n*=5 fins), and is found at the tips of double mutant rays (white asterisk), not forming fibers (J) (*n*=4 fins). Scale bars: 50 µm.

with wild-type females performed all the above behaviors in succession, with successful egg release (Movie 1; Fig. 5B). The double mutant males attempted to Wrap-around the female, shown by executed Freezing and Hooking behavior (Fig. 6B; Movie 1, sequence 1). However, no eggs were released. Another courtship sequence showed multiple failed Wrap-around attempts between a double mutant male and wild-type female (Movie 1, sequence 2). Overhead views showed the male hooking his trunk around the female; however, he was unable to grasp her (Movie 1, sequence 2). These results suggest that a proper male pectoral fin phenotype is important for stimulation of egg release and/or for providing the necessary grip to grasp the female.

It has been suggested that stimulation of egg release is facilitated by sexually dimorphic keratinized epidermal structures, called breeding tubercles (BTs), located on the dorsal surface of male pectoral fins (Kang et al., 2013; Wiley and Collette, 1970; Kortet et al., 2004; McMillan et al., 2013). In zebrafish, BTs are arranged in clusters along the anterior-central male pectoral fin rays (Fig. 6C; Fig. S9D) (Kang et al., 2013; McMillan et al., 2013; Fischer et al., 2014). BT clusters were present on these rays in double mutant males, with abnormalities. Although, occasionally, they appeared comparable to those of wild-type males (11/44 rays with BTs), BTs within the clusters were often reduced in size (25/44 rays with BTs) or formed shorter interrupted clusters (8/44 rays with BTs) when compared to wild-type siblings of the same age, SL, and sexual experience (Fig. 6C). Mallory's trichrome stain (Cason, 1950) of transverse cryosections of both wild-type and double mutant BT clusters showed a hollow keratinized cap stained red, supported by a raised epidermis stained blue (Fig. 6D), suggesting that wild-type and double mutant BTs were similar at the tissue level. However, the number of rows of keratinized spikes varied (Fig. 6D). These abnormalities may be responsible for a fin phenotype that is not appropriate for stimulating or grasping females. Individual BTs are also found regularly spaced along the length of the distal half of the dorsal, anal and pelvic fin rays of wild-type males (Fig. 6E) (McMillan et al., 2013). The distribution of BTs in these fins was abnormal in double mutant males (Fig. 6E). BTs are also found on the lower jaw, another area of contact between male and female zebrafish (Wiley and Collette, 1970; Kortet et al., 2004; McMillan et al., 2013). Whole-mount Mallory staining of dissected lower jaws revealed intact BTs in both wild-type and double mutant males (Fig. 6E). This suggests that the abnormalities observed in double homozygous male BTs are fin specific, and may therefore be related to the ray defects conferred by the loss of actinotrichia. The disturbed BT phenotype in double mutants may be a consequence of their malformed rays and may impair the execution of successful courtship behavior. Therefore, actinotrichia play an indirect role in zebrafish reproduction by promoting a pectoral fin ray morphology necessary for properly patterned pectoral BT clusters.

## DISCUSSION
We generated an *and1$^{-/-}$and2$^{-/-}$* zebrafish line, carrying partial deletions in the *and1* and *and2* genes, to disrupt actinotrichia synthesis and examine the impact of actinotrichia loss in fin development. We demonstrate that And genes are necessary for the synthesis of actinotrichia. The lack of actinotrichia leads to defects in FF morphology and in the organization of FF mesenchyme migration, culminating to the malformation of the adult fins. This is consistent with our previous hypothesis, where we suggested that the loss-of-function of And genes would lead to adult fin defects (Zhang et al., 2010). We demonstrate that the actinotrichia are crucial for the proper patterning of all skeletal structures developing within the FF,

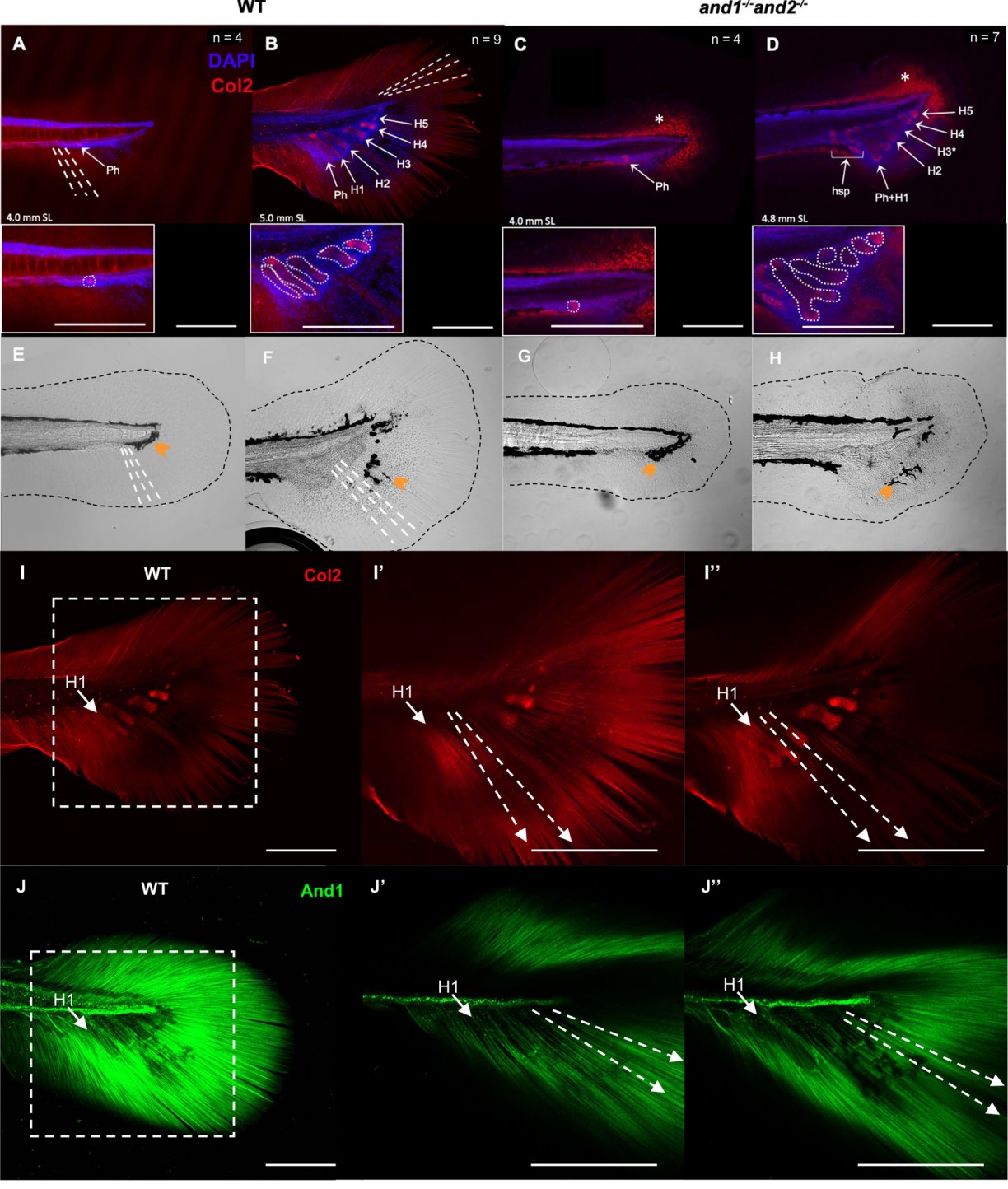

**Fig. 5. Hypural fusions occur during caudal fin development.** (A-D) Maximum intensity projection confocal images of Col2-immunostained whole-mount wild-type sibling (A,B) and double mutant (C,D) median fin folds (FFs). Actinotrichia (white dashed lines) proximodistally span the wild-type FFs. Col2 is aggregated at the base of double mutant FFs (white asterisks). Hypural elements are indicated and outlined (white dotted lines) in insets. Ph, parhypural; H, hypural. Scale bars: 200 μm. (E-H) Transmitted light differential interference (TD) images of A-D. Black dotted lines outline fin margins. Melanocytes (orange arrowheads) are deposited around the notochord (E,G) and migrate distally as the rays develop (F,H). Actinotrichia (dashed white lines) are seen in wild-type FFs (E,F). (I) Maximum intensity projection of B, showing only Col2 signal (also shown in Fig S8C WT1). (I′,I″) Magnified slices from a z-stack showing the region outlined in I at depths above (I′) and below (I″) the hypurals. Hypural 1 (H1) is labelled for reference. White dashed arrows indicate the actinotrichia. Hypurals reside among actinotrichia and seem to grow in the same plane (I′,I″). Scale bars: 200 μm. (J) Maximum intensity projection of And1-immunostained whole-mount 5.2 mm standard length (SL) wild-type sibling median FF (n=4 fish). (J′,J″) Magnified slices from a z-stack showing the region outlined in J at depths above (J′) and below (J″) the hypurals. Hypural 1 (H1) is labelled for reference. White dashed arrows indicate the actinotrichia. Hypurals are developing between two arrays of actinotrichia situated above and below the cartilages. n=4 fish in A,C,E,G,J; n=9 fish in B,F,I; n=7 fish in D,H. Scale bars: 200 μm.

which include the rays of all fins and the caudal fin endoskeleton. Additionally, we found that actinotrichia are also crucial for spawning success in zebrafish through their role in the proper patterning of the male pectoral rays and associated BT clusters. We therefore propose that And genes are also indirectly involved in the maintenance of functional BT cluster morphology and reproductive success, and therefore overall organism reproductive fitness.

## Actinodin genes are required for actinotrichia synthesis

In addition to And proteins, collagen proteins are also important structural constituents of actinotrichia. Functional analysis of Collagen1a1, Collagen2a1b and Collagen9a1c proteins, and their contributions to actinotrichia has previously been analyzed (Fisher et al., 2003; Durán et al., 2011; Nakagawa et al., 2022). Morpholino-mediated knockdown of *col1a1* and *col2a1b* results in the formation of disorganized, but still present, actinotrichia (Durán et al., 2011). The *col1a1$^{dc124}$* mutation results in atypical Collagen1a1 molecules and has been linked to skeletal dysplasia (Fisher et al., 2003). Like the *and1$^{-/-}$and2$^{-/-}$* mutants, these collagen mutants present with truncated fins, wider rays with reduced bifurcation and absent actinotrichia; however, they can also form phenotypically normal rays with actinotrichia (Durán et al., 2011). *col9a1c* mutation also caused truncated and malformed fin rays, but these mutants retain their actinotrichia and only develop defects in fin morphology at later stages in development (Nakagawa et al., 2022; Huang et al., 2009). These loss-of-function analyses show that the collagenous component of actinotrichia is necessary for proper fin formation, demonstrated through fin fold and fin defects that are similar to those of our double mutants. However, perturbing the synthesis of actinotrichia collagens does not prevent the fibers from forming, in contrast to And component disruption. These analyses perturbed the function of one collagen gene at a time; therefore, it is possible that combining multiple mutations in collagen genes may result in the loss of actinotrichia. We show that the contribution of both *and1* and *and2* to the non-collagenous Actinodin component of elastoidin is necessary for actinotrichia synthesis. Partial deletions of *and1* and *and2* result in absent or severely reduced expression, likely due to nonsense-mediated mRNA decay (Wittkopp et al., 2009). The simultaneous loss of function of *and1* and *and2* resulted in the absence of actinotrichia from the FFs, even while the shorter paralogous pair, *and3* and *and4*, were intact. These latter two are also expressed in the FFs during development (Zhang et al., 2010), but encode much shorter proteins with fewer functionally important structural repeats (Zhang et al., 2010). Interestingly, we did notice a significant upregulation of *and4* at 5 dpf in double mutants relative to wild-type siblings (Fig. 1C). However, the normalized expression levels of both *and3* and *and4* remain low in double mutants and wild-type siblings (Fig. S3B), so any compensatory upregulation is insufficient to rescue the defects caused by *and1* and *and2* disruption. Moreover, this upregulation seems temporary, as there is no upregulation of *and3* and *and4* later in development, at 14 dpf (Fig. S8D,E). Genes encoding type II collagen, *col2a1a* and *col2a1b*, are not significantly different, suggesting no compensation, at least in larval stages (Fig. 1C; Figs S3B and S8D,E). We therefore suggest that *and1* and *and2* are major contributors to actinotrichia synthesis, with *and3* and *and4* playing a less consequential role.

Our double mutants also show that the absence of actinotrichia causes disorganized migration of the FF mesenchymal cells, which are known to contribute to ray formation (Nakamura et al., 2016; Lee et al., 2013). This disorganization contributes to the malformation of the rays, including their shape, their irregular joint spacing and their

reduced bifurcation, and leads to the formation of fewer rays. This supports our previous hypothesis that the loss of the And gene family in land-adapting fish was an important step in the fin-to-limb transition (Zhang et al., 2010). Indeed, reduced And gene expression has been noted in the leg-like appendages derived from the pectoral fins in the sea robin (*Prionotus carolinus*) (Herbert et al., 2024). We demonstrate that And gene loss of function leads to a reduction in the fin dermoskeleton, a pattern that has been reported in the fossil record with the evolution of limbs from paired fins (Shubin et al., 2006; Stewart et al., 2020; Coates, 1994). Shorter pectoral fin rays have also been reported in *and1/and2* crispants (Kudoh et al., 2024). Our findings, combined with the absence of And genes from tetrapods, lead us to propose a scenario where the loss of the And gene family or the loss of And gene expression would have led to the loss of actinotrichia in the developing fin, resulting in the reduction of the fin dermoskeleton.

## Actinotrichia are important for the development of fin fold-localized skeletal elements

In addition to fin dermoskeleton defects, the double mutants have fusions of the endochondral hypural elements in the caudal fin. This was unexpected because, in the adult caudal fin, the actinotrichia are found at the distal extremities of the rays, while the endoskeleton is at the proximal base of the fin. However, analysis of hypural formation shows that they develop between the two arrays of actinotrichia within the median FF, and therefore their patterning may be influenced by the actinotrichia, like the rays. While it is known that the FF mesenchyme contributes to the fibroblast and osteoblast lineages that form the dermoskeleton (Nakamura et al., 2016; Lee et al., 2013), it may also contribute to condensations forming endochondral bones, such as the hypural elements. The hypurals develop by endochondral ossification, where ventroposterior FF mesenchymal cell condensation gives rise to chondrocytes that direct adjacent perichondral cells to differentiate into osteoblasts (Anderson et al., 2020; Papaioannou et al., 2015). The hypural elements seem to grow in the same plane and general angle as the actinotrichia (Fig. 5I,J). This suggests that the presence of actinotrichia during caudal fin endoskeleton development influences the patterning of associated chondrocytes, in addition to the fibroblasts and the osteoblasts of the rays. Extracellular scaffolds, like actinotrichia, can provide anchoring sites for chondrocytes (Aigner and Stöve, 2003). Col2a1 itself can act as a signaling molecule in chondrocyte metabolism (Garamszegi et al., 2010; Xin et al., 2015), acting through chondrocyte integrin receptors for Col2a1 (Enomoto et al., 1993). Actinotrichia, containing fibrillar Col2a1, could therefore have a direct role in chondrocyte behavior in the developing hypurals. However, actinotrichia may also simply provide structural support and spatial restriction for chondrogenesis. Although actinotrichia and the hypural cartilages overlap spatially in wild-type larvae (Fig. 5I,J), in the absence of actinotrichia, Col2 proteins aggregate within the FF, away from the hypurals developing on the ventral side of the notochord, with no spatial overlap (Fig. S8C). There was also no significant upregulation of *col2a1a* and *col2a1b* prior to (Fig. 1C) or during (Fig. S8D) hypural development. This suggests that the aggregates contain Col2 that would otherwise contribute to actinotrichia in the presence of actinodin, but Col2 aggregates do not contribute to hypural fusions. Fusions of the endoskeleton were only present in the caudal fin because the hypurals are the only skeletal elements developing within the FF. This supports a role for actinotrichia in influencing the patterning of those skeletal structures that develop within the FF.

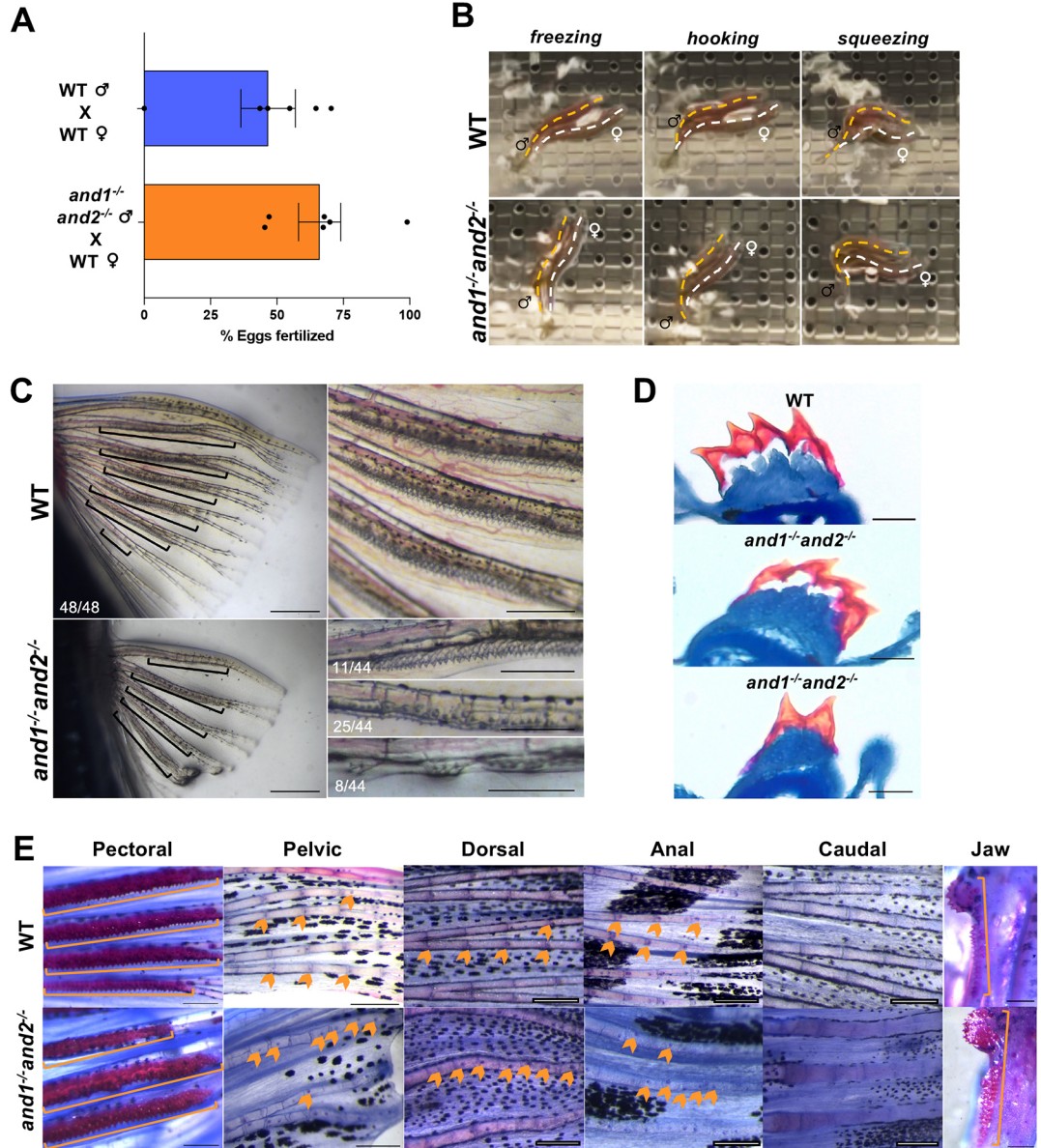

**Fig. 6. Actinotrichia contribute to the pectoral breeding tubercle cluster phenotype required for successful zebrafish reproduction.** (A) Percentage of wild-type eggs (*n*=6 females) fertilized by wild-type (*n*=3 males) and double homozygous (*n*=3 males) sperm by IVF. Data pooled from two trials of 3 wild-type male×wild-type female and 3 *and1*$^{-/-}$*and*$^{-/-}$ male×wild-type female crosses. Data are mean±s.e.m. Dots represent individual data points. (B) Stills from overhead views of courtship videos of wild-type and double mutant males with wild-type females in Movie 1. Double mutant stills were taken from courtship sequence 1. Body contortions are emphasized at dorsal midlines (males, orange dashed lines; females, white dashed lines). (C) Pectoral breeding tubercle cluster phenotype in wild-type [*n*=48 rays with breeding tubercles (BTs)] and double mutant (*n*=44 rays with BTs) males. Left: whole pectoral fins show the arrangement of BT clusters relative to ray phenotype. Brackets indicate BT clusters. Scale bars: 1 mm. Right: magnified views of the whole fin. Wild-type BT clusters are found on anterior-central pectoral fin rays. Double homozygous BT clusters present with a range of defects, appearing normal (*n*=11/44), reduced (*n*=25/44) or segmented (*n*=8/44) in different fin rays of different individuals. Scale bars: 500 μm. (D) Mallory-stained transverse cryosections of wild-type (*n*=32 sections, 3 males) and double homozygous (*n*=30 sections, 3 males) pectoral BT clusters. Double mutant clusters contained 3 or 4 rows of BTs (*n*=12/30), comparable to wild-type clusters, but also often had 1 or 2 rows of BTs (*n*=18/30). Scale bars: 100 μm. (E) Whole-mount Mallory staining of all fins and jaw in wild-type (*n*=4) and double mutant (*n*=3) males. BTs found along the dorsal, anal and pelvic fin rays are stained darker than their periphery. BTs arranged in dense clusters in the pectoral fin and the jaw are stained red. Orange arrowheads indicate individual BTs; orange brackets indicate BT clusters. Scale bars: 250 μm.

## The relationship between actinotrichia and fish courtship strategies

Double mutant males are unable to stimulate female egg release during courtship, which we generally attribute to their pectoral fin ray defects. The importance of male pectoral fins in zebrafish courtship has previously been described; Kang et al. (2013) showed, using amputations, that intact male pectoral fins are required for successful spawning. Zempo et al. (2021) demonstrated that males physically use their pectoral fins to stimulate egg release in females. This coincides with our findings that double mutant males, which have pectoral fin defects, are unable to stimulate egg release. Although it is possible that the reduced fin size and ray defects prevent males from stimulating female egg release, the double mutant males also have abnormal pectoral fin BTs, which have also been posited to be

involved in reproduction in other fish (Wiley and Collette, 1970; Kortet et al., 2004, 2003; Yamamoto and Egami, 1974). Pectoral BTs form at the onset of puberty (Dai et al., 2021) on already-developed rays, which, in the double mutants, are malformed and could impair BT cluster morphology indirectly. Further, some double mutant males still appear to grasp females (Fig. 6B; Movie 1), suggesting that egg release depends on an input additional to squeezing, such as physical stimulation by pectoral BTs.

Regardless, the inability of the double mutant males to breed has drastic fitness consequences, as they cannot generate offspring. As such, the loss of And genes would be extremely disadvantageous in fish utilizing similar mating strategies to zebrafish, suggesting a strong selection for the maintenance of actinotrichia for reproductive success. Epidermal tubercles have been observed in ancient fish fossils, and are thought to be morphologically and functionally similar to those that are used by extant fish for breeding (Xu and Zhao, 2016; Lombardo, 1999). For example, well-preserved Triassic specimen of the sexually dimorphic *Venusichthys comptus* show BT-like structures on cranial structures, scales and fins of presumed males (Xu and Zhao, 2016). Many small Triassic ray-finned fish are presumed to be viviparous, based on the presence of hooks on specialized anal fins of presumed males, which appear similar to those of extant viviparous teleosts and are involved in sperm transfer. However, presumed *V. comptus* males lack such a specialized anal fin, suggesting that they adopted a different reproductive strategy and may not have been viviparous (Xu and Zhao, 2016). It is possible that courtship behavior similar to that of zebrafish was utilized in ancient aquatic fish. Although *V. comptus* occupies the more derived Neopterygian clade within Osteichthyes, it raises the possibility that a basal Osteichthyan fish living prior to the divergence of ray-finned fish and lobe-finned fish may have opted for a courtship mechanism involving BTs for external fertilization. Our findings suggest that fish using this strategy would experience a strong selective pressure to maintain their actinotrichia for proper patterning of their fin ray-associated BTs. We speculate that emergence of tetrapods may involve a shift in courtship behavior, in addition to the loss of And genes. However, the order of these events is unknown. Zebrafish also have a relatively derived fin morphology (Coates, 1994; Coates and Cohn, 1998; Grandel and Schulte-Merker, 1998; Lalonde and Akimenko, 2018) and occupy a derived phylogenetic position relative to the fish ancestral to tetrapods. As such, zebrafish studies will benefit from complementary comparative analyses.

In conclusion, we show that the loss of function of the *and1* and *and2* genes in zebrafish prevents the synthesis of actinotrichia fibers that are crucial for proper fin development. This results in morphological defects of the median and pectoral fin folds, which accumulate to cause malformed adult fins. These defects have an especially pronounced effect in the male pectoral fins, as they seem to impair the patterning of the breeding tubercles that develop on the rays, which are used to stimulate egg release in females. We propose that actinotrichia are not only important for proper fin development, but also in reproductive success in zebrafish and other cypriniforms using a pectoral fin and/or BT-mediated courtship. As such, we speculate that this role of actinotrichia in fish reproduction applied a strong pressure to maintain the And genes in aquatic fish that utilized this BT-mediated courtship strategy. The loss of the And gene family during tetrapod evolution may have therefore occurred with a shift in courtship strategy independent of stimulation by pectoral breeding tubercles or other action from the fins.

## MATERIALS AND METHODS
### Animals
All experiments were conducted at the University of Ottawa zebrafish (*Danio rerio*) facility in compliance with guidelines and protocols for animal use

approved and certified by the Canadian Council on Animal Care and licensed under the Ontario Animals Research Act. Adults and embryos were maintained as previously described (Westerfield, 2000). Adult zebrafish were anesthetized using 0.2 mg/ml tricaine methanesulfonate (Sigma-Aldrich). Euthanized specimens were fixed in 4% paraformaldehyde (PFA) and stored in 100% methanol at −20°C until use. The *and1/and2* mutants were generated from a wild-type stock colony maintained at the University of Ottawa zebrafish facility. The ET37 transgenic line, kindly supplied by Vladimir Korzh (Institute of Molecular and Cell Biology, Singapore) (Choo et al., 2006), was crossed with the *and1/and2* double mutants to generate ET37⁺ double mutants and wild-type siblings expressing GFP in their fin fold mesenchymal cells.

### Generation of *and1* and *and2* mutations using the CRISPR-Cas9 system
The CRISPR/Cas9 system was used to generate large deletions in the *and1*- and *and2*-coding regions. Target sites and corresponding sgRNAs were selected using CRISPRscan (Moreno-Mateos et al., 2015) (*and1*, 5′-CCTCTCTTTAGAGGTTCACACCA-3′ and 5′-AGCACTCTGACTTCTG-GAATTGG-3′; *and2*, 5′-CTGAAAGCACCTCTAGCCCA-3′ and 5′-GTA-ATCCCGCCACTGACCCT-3′). Cas9 was transcribed using an mMessage mMachine Sp6 kit (Invitrogen) from the pCS2-nCas9n plasmid (Addgene #47929), linearized with NotI. The injection mix contained 300 ng/µl Cas9 mRNA and 12.5 ng/µl sgRNAs. Primary injected (F0) fish were raised to adulthood and crossed with wild-type adults. Fish were genotyped using the following primers: *and1* FW, 5′-GGGGTTTGGCTCACCTTTGTC-3′; *and1* RV1, 5′-GCACTTGAGTTTTGCTGACAACAAG-3′; *and2* FW, 5′-GGT-GCCAATTTCAACCATTACCTC-3′; *and2* RV1, 5′-CGGATCCACCTAT-CAGACAATAGG-3′. Single heterozygous mutants were inbred to establish single homozygous lines, which were selectively bred to generate double heterozygous and finally double homozygous lines.

### *In situ* hybridization
Digoxigenin-labelled antisense RNA probes were prepared from total 2 dpf cDNA as previously described (Hua et al., 2018), using the following primers: *and1* FW, 5′ TGTTCTATTCTTGTCATACCTGCAG 3′; *and1* RV, 5′ CTT-GAACATGGCATAAGGATC 3′ (822 bp amplicon); *and2* FW, 5′ CCACT-GACCCTCTTTGTGT 3′; *and2* RV, 5′ TCATTTTCTTGTAGCCACCC 3′ (689 bp amplicon). The amplicons were re-amplified using the additional primers: T7 *and1* RV, 5′ CAGTGAATTGTTAATACGACTTCACTTATA-GGGCTTGAACATGGCATAAGGATC 3′; T7 *and2* RV, 5′ CAGTGAAT-TGTTAATACGACTTCACTTATAGGGTCATTTTCTTGTAGCCACCC 3′. Antisense probes were transcribed *in vitro* using T7 RNA polymerase. *In situ* hybridization was performed as previously described on whole-mount embryos (Thisse and Thisse, 2008), with a 30-min digestion in 1:1000 10 mg/ml proteinase K, and on 16 µm cryosections (Smith et al., 2008).

### RT-qPCR
RNA from three wild-type and three double mutant pools of 30 5 dpf larval tails or 15 14 dpf larval tails, cut just posterior to the urogenital pore, was extracted by sonication in TRIzol reagent (Invitrogen), following the manufacturer's protocol. RNA purity was validated by a NanoDrop One Microvolume UV-Vis Spectrophotometer (Thermo Scientific, 840274100). cDNA was synthesized using the iScript Reverse Transcription Supermix for RT-qPCR kit (Bio-Rad), according to manufacturer protocol. For all genes, 10 µl reactions were prepared as follows: 5 µl SYBR Green for qPCR (Bio-Rad), 0.4 µl forward primer (10 µM), 0.4 µl reverse primer (10 µM), 4 µl template cDNA (or nuclease-free water for no-template controls) and 0.2 µl of nuclease-free water. The following primers were used: *and1* (Gene ID 794315) FW, 5′ CGTGACCCAGCGTTTAAAAG 3′; *and1* RV, 5′ AGAGTGCTGCTTTGTACCAACTG 3′; *and2* (Gene ID 566702) FW, 5′ AGTGCTGGTCCTGTAAAGCG 3′, *and2* RV, 5′ CCCTCCTGGTTGTTGGTTCCTC 3′; *and3* (Gene ID 561047) FW, 5′ AGTAACAGGATGGCAGACGG 3′; *and3* RV, 5′ GCAAGAGATGTGG-CCTGTG 3′; *and4* (Gene ID 100192207) FW, 5′ AGGTCATGATGTC-TGCGGT 3′; *and4* RV, 5′ TCCGAGTACTCAAAGACAGGTC 3′; *col2a1a* (Gene ID 562496) FW, 5′ TCTACTCGATCACAGTCTTGGC 3′; *col2a1a* RV, 5′ AAGAAGGCCATTTCTGCTGC 3′; *col2a1b* (Gene ID 503730) FW, 5′ AAGACCTGGAGACGCTGGT 3′; *col2a1b* RV,

5′ CTGGAAGACCAACGATACCG 3′; *ef1a* (Gene ID 30516) FW, 5′ CTGGAAGGCCAGCTCAAACAT 3′; *ef1a* RV, 5′ ATCAAGAAGAG-TAGTACCGCTAGCATTAC 3′; *rpl13a* (Gene ID 560828) FW, 5′ TCTGGAGGACTGTAAGAGGTATGC 3′; *rpl13a* RV, 5′ AGACGCA-CAATCTTGAGAGCAG 3′. Three technical replicates were prepared for each biological replicate. Reactions were carried out using the Bio-Rad CFX96 system. Cq values were normalized against two reference genes, elongation factor 1 alpha (*ef1a*) and ribosomal protein l13a (*rpl13a*). Fold changes were analyzed using the ΔΔCq method (Livak and Schmittgen, 2001).

## Fluorescent immunohistochemistry

For whole-mount immunohistochemistry, fixed adult fin tissue was depigmented in a solution of SSC, formamide and hydrogen peroxide. Immunostaining was performed as previously described (Smith et al., 2008), with an additional antigen retrieval wash in 0.01 M sodium citrate buffer (pH 6), and with a 30-min permeabilization in a 1:1000 dilution of 10 mg/ml proteinase K. A blocking solution of 10% calf serum in 1×PBS was used. For And1 immunohistochemistry, a custom-made rabbit anti-And1 primary antibody (1:1000, CGQDDHLAYNGDYRKK, Genscript), previously validated by König et al. (2018), and a goat anti-rabbit secondary antibody (1:500, Invitrogen, A-11008) were used. For Col2 immunohistochemistry, a mouse anti-Col2 primary antibody (1:500, II-II6B3-s, Developmental Studies Hybridoma Bank, AB_528165) and goat anti-mouse Alexa Fluor 594 secondary antibody (1:500, Invitrogen, A-21207) were used.

## Histology

Mallory's trichrome stain on 16 μm cryosections and whole-mount tissue was performed as previously described (Cason, 1950), with the following modifications: tissue was fixed in 4% PFA, tissue for sections was embedded in one-third 30% sucrose in PBS with two-thirds OCT cryo embedding matrix (Fisher Scientific) and mounted in DPX mounting medium (Sigma); clearing and mounting steps were omitted for whole-mount tissue. Sections were imaged using a Zeiss Axio Zoom.V16 microscope. Whole-fish skeletal staining was performed as previously described (Sakata-Haga et al., 2018). Masson's Trichrome staining of 4 μm paraffin wax-embedded sections was performed by the Louise Pelletier Histology Core Facility (RRID: SCR_021737) at the University of Ottawa.

## Imaging

Bright-field images of whole embryos, fins and sections were taken using a Zeiss AxioZoom.V16 microscope. Dissected FFs were imaged using a Nikon A1R confocal microscope. Confocal images were captured using Olympus Fluoview FV1000 or Nikon A1R microscopes. For live imaging of fins, adults were anesthetized in tricaine (0.2 mg/ml) and imaged using a Leica MZFLIII stereoscope. Whole live fish were imaged using a 12MP camera. All images were processed in Image J Fiji.

## Measurements and statistical analyses

The lengths and widths for all adult fins, pectoral FFs, endoskeletal discs and median FFs were measured using Image J. For the caudal fin, the dorsal lobe width and the median cleft length were also measured (Fig. S5B). Measurement schematics are indicated by schematics corresponding to the appropriate graph. Measurements were normalized to the SL of each fish to account for variation in body size at a given age. Graphs were made using GraphPad Prism (version 10.2.3). Box plots show means (+), first and third quartiles (box), and lowest and highest values (whiskers). Dots indicate individual values. Statistical analysis details are noted in the figure legends corresponding to each graph. Two-tailed unpaired nonparametric Mann–Whitney $u$-tests ($\alpha$=0.05) were used to compare wild-type and double mutant measurements, with an FDR correction of 1% to account for multiple comparisons. RT-qPCR analysis was performed using the ΔΔCq method (Livak and Schmittgen, 2001), with a two-tailed unpaired $t$-test ($\alpha$=0.05) to determine significance.

## Courtship behavior video capture

Age- and size-matched male and female zebrafish were housed in 1:1 pairs for the 3-week duration of the behavior experiments. The evening prior to video capture, each pair was placed in a breeding tank, separated by a divider. The divider was removed the next morning at daybreak, and behavior was recorded for 10 min on a 12MP f/1.8 aperture camera at 1080p and 60 frames per second. Videos were annotated using Lightworks v2023.1. One breeding session per week was conducted.

## *In vitro* fertilization

Males and females were housed in breeding tanks, separated by a divider, the evening prior to experiments. *In vitro* fertilization was performed as previously described (Westerfield, 2000). Milt was extracted from anesthetized males by gentle pressing on their abdomen and collected in 100 μl of Hank's solution (Sigma-Aldrich). The milt solution was poured over extracted eggs. Fertilized embryos surviving to 10 hpf were counted over the total number of eggs extracted from each female. One trial per month was conducted.

### Acknowledgements

We thank Dr Marc Ekker for critical reading of the manuscript. We gratefully acknowledge the histology service provided by the Louise Pelletier HCF at the University of Ottawa.

### Competing interests

The authors declare no competing or financial interests.

### Author contributions

Conceptualization: P.H., M.-A.A.; Methodology: P.H., C.B., B.K., R.K., R.L.L., M.-A.A.; Validation: P.H., C.B., B.K., R.K., R.L.L., M.-A.A.; Formal analysis: P.H., C.B.; Investigation: P.H., C.B., B.K., R.K., R.L.L.; Resources: R.L.; Data curation: P.H., C.B., B.K., R.K., R.L.L.; Writing – original draft: P.H., M.-A.A.; Writing – review & editing: P.H., C.B., B.K., R.K., R.L.L., M.-A.A.; Visualization: P.H.; Supervision: M.-A.A.; Project administration: M.-A.A.; Funding acquisition: M.-A.A.

### Funding

This work was supported with grants awarded to M.-A.A. from the Natural Sciences and Engineering Research Council of Canada (RGPIN-2024-06801) and the Canadian Institutes of Health Research (PJT 166139). Open Access funding provided by University of Ottawa. Deposited in PMC for immediate release.

### Data and resource availability

All relevant data and details of resources can be found within the article and its supplementary information.

### Peer review history

The peer review history is available online at https://journals.biologists.com/dev/lookup/doi/10.1242/dev.204990.reviewer-comments.pdf

### Special Issue

This article is part of the Special Issue 'The Extracellular Environment in Development, Regeneration and Stem Cells', edited by Alex Hughes and Rashmi Priya. See related articles at https://journals.biologists.com/dev/issue/153/16

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
