## [Peer Review File · Development (Cambridge, England)]

An essential role for actinotrichia in zebrafish fin patterning and courtship behavior

Paulina Hanzelova, Connor Baird, Bidemi Keshinro, Reeham Kadhom, Robert Lalonde and Marie-Andrée Akimenko
DOI: 10.1242/dev.204990

Editor: Steve Wilson

Review timeline

Original submission:	30 May 2025
Editorial decision:	26 June 2025
First revision received:	8 October 2025
Editorial decision:	6 November 2025
Second revision received:	26 February 2026
Accepted:	16 March 2026

Original submission

First decision letter

MS ID#: dev.204990

MS TITLE: An Essential Role for Actinotrichia in Zebrafish Fin Patterning and Courtship Behaviour

AUTHORS: Paulina Hanzelova, Connor Baird, Bidemi Keshinro, Reeham Kadhom, Robert Lalonde and Marie-Andree Akimenko

Dear Marie-Andree,

I have now received all the referees' reports on the above manuscript, and have reached a decision. The referees' comments are appended below, or you can access them online: please go to:

As you will see, the referees express considerable interest in your work, but have some significant criticisms and suggestions for improving your manuscript. If you are able to revise the manuscript along the lines suggested, I will be happy receive a revised version of the manuscript. Please also note that Development will normally permit only one round of major revision. If it would be helpful, you are welcome to contact us to discuss your revision in greater detail. Please send us a point-by-point response indicating your plans for addressing the referees' comments, and we will look over this and provide further guidance.

Please attend to all of the reviewers' comments and ensure that you clearly highlight all changes made in the revised manuscript. Please avoid using 'Tracked changes' in Word files as these are lost in PDF conversion. I should be grateful if you would also provide a point-by-point response detailing how you have dealt with the points raised by the reviewers in the 'Response to Reviewers' box. If you do not agree with any of their criticisms or suggestions please explain clearly why this is so.

Reviewer 1

Advance summary and potential significance to field

In a previous study, the authors had shown that morpholino-mediated co-knockdown of the actinodin genes *and1* and *and2*, which form the structural basis of actinotrichia in zebrafish fins, led to temporary defects in fin fold formation and cell migration. The aim of this study was to examine the roles of *and1* and *and2* during the development of zebrafish larval and adult fins. To this end double-mutant fish were generated by inducing loss-of-function *and1*- and *and2*-alleles with CRISPR/Cas9 methods. Double-mutant larvae lacked evidence of actinotrichia fibres and fin fold mesenchyme migration was perturbed. Double-mutant adults present overall smaller fins with partially disformed fin rays. The cartilaginous templates for hypural bones in the caudal fin were disorganized, leading to partial fusions. This suggests that actinotrichia supply a structural scaffold in the fin fold for the caudal fin endoskeleton. The importance of breeding tubercles for zebrafish courtship success had been noted previously. Because *and1/2*^{-/-} males exhibit improperly developed breeding tubercles on their pectoral fins (while tubercles developed normally elsewhere), they fail to elicit egg release during courtship behavior. The authors show that actinodin genes are important for adult teleost fin development, and discover that actinotrichia in the fin fold guide the formation of cartilaginous precursors of the hypural bones of the caudal fin endoskeleton. They propose an evolutionary scenario in which the loss of actinodin genes in tetrapods was accompanied by a reduction of the fin dermoskeleton. This manuscript addresses interesting questions for the community interested in limb development, the role of actinodins in fin development beyond the embryo. Unexpectedly, it also reveals that actinodins are required in the fin fold to guide developmental processes in the cartilaginous precursors to the adult caudal fin endoskeleton. The finding that male double-mutants are infertile is curious, but to infer that the evolution of land tetrapods might have necessitated a switch in courtship procedures might overstretch its importance. We do not know if tubercle-like structures on fossils basal to the tetrapod lineage even exist, nor do we know if they were used as in zebrafish. Nevertheless, this is carefully controlled and thorough study.

Comments for the author

Specific comments:

- 1) In Fig.1E it is not easily to see what the authors mean. Elongation and arborization should be marked in a clear way.
- 2) Lines 207-209: Extension of rays past the distal extremity of the fin and reduced interray tissue are basically the same thing, no? Or is the idea that two independent processes are affected in double-mutants?

Comments that need to be addressed in the discussion:

- 1) The downregulation of *and* transcription in the mutants is curious. How do the authors interpret this finding? Is this an autoregulatory feature that can be overcome by injection of *and* mRNA? Are the remaining *and* genes not expressed in fins?
- 2) There is a triassic fossil, *V. comptus*, that has BT-like structures on its fins, suggesting that this type of courtship may have already been utilized by some ancient fish. However, this fossil is a member of the derived Neopterygian class and thus not a direct precursor of the lobe-finned fish from which the tetrapods derived. I would propose to tone down the claim that a shift in courtship strategy was required for the fin-to-limb transition (as highlighted in the summary statement).
- 3) If the idea was true that *and* genes are essential for courtship behavior, it might be interesting to examine the available fish genomes for the presence or absence of *and* genes. If there are examples where a loss has happened, it would be important to learn about their courtship rituals and fin development. Along these lines, it would be useful to speculate why either *and1* or *and2*, although redundant in zebrafish fins, have not been eliminated by evolutionary processes.
- 4) Typo in line 296 (hadd).

Reviewer 2

Advance summary and potential significance to field

This study characterizes the phenotype of double mutant zebrafish with deletions of actinodin 1 and actinodin 2 (*and1/2*) genes. Actinodin proteins are fish specific as they are essential for actinotrichia, which are extracellular fibrils supporting the larval fin fold and the distal end of the adult caudal fin. The same laboratory has previously reported that morpholino knockdown of

and1/2 impaired fin fold formation. The goal of this study is to determine the post-embryonic phenotype using genetic approach. The study reports a lack of actinotrichia in double mutants. This loss led to reduced fins both in larvae and adults. Males were unable to stimulate females to lay eggs during the courtship. The authors conclude that actinodin genes are essential for fin development and reproductive success. The authors interpreted these finding in the context of evolutionary fin-to-limb transition, because tetrapods lack actinodin genes and actinotrichia in their limbs.

The characterization of the and1/2 mutants is novel and highly relevant in skeletal biology of fish. The data expand our knowledge on the role extracellular matrix in development of the fins, which are essential not only for locomotion but also for mating in zebrafish. Nevertheless, some limitations can be recognized. Although the conclusions of the abstract and the summary sound compelling, the observed phenotype is less severe than expected, based on literature. All the fins are formed in the double mutant, but they are smaller, and the rays are less regular than in wild type. In addition, some fusions of a few hypural bones in caudal endoskeleton occur, which, however, do not impact the swimming behaviour, as recorded on videos. The summary statement needs to be toned down to accurately reflect the reported findings.

The direct versus indirect phenotypes should be carefully defined to improve clarity and to avoid a risk of misinterpretation.

The impact of the paper could be improved if some molecular mechanisms were included in the study, following the phenotypic characterization of the mutants.

The significance of the paper would potentially benefit by assessing not only fin development but also fin regeneration in the and1/2 mutants.

Comments for the author

Major specific points

1. How do actinotrichia look like?

Actinotrichia is the subject of this study.

Many readers of Development journal are not experts in skeletal biology of fish, being not yet familiar with these peculiar fibers. Thus, it would be helpful to start with an illustration to introduce the structure and distribution of actinotrichia.

2. Where are actinotrichia in Figure 1?

Line 127 / Fig. 1C

The main conclusion of the paper is that the and1/2 double mutants lack actinotrichia. However, the data for this statement are not convincing, because actinotrichia are not visualized in the specimens.

Actinotrichia should be distinctively labeled to conclude about their presence or absence. On the image, 3 red lines seemingly depict actinotrichia. How have they been tracked on bright-field photographs of live zebrafish larvae? How many actinotrichia are expected in wild type embryos? If we cannot well see an actinotrichium in wild type, how can we see its absence in mutants?

To provide any evidence about these structures, it is necessary to perform histological staining for collagenous bundles of actinotrichia.

3. Do double mutants really lack any form of actinotrichia?

Beside actinodin 1 and 2, there is also actinodin 3 and 4, as mentioned in the introduction, which could contribute to actinotrichia.

Is it possible that the size and distribution of actinotrichia depend on the composition, specifically on the type of actinodins? If yes, shorter or thinner actinotrichia would be less conspicuous, and they could easily be overlooked in bright field or fluorescence images.

To support the main conclusion, it would be necessary to conduct ultrastructural analysis to prove the absence of any actinotrichia in the double mutant fin.

4. How severe and reproducible is the phenotype with the irregular margin of the fin and wrinkling of the tissue?

Line129 Fig. 1C

The described phenotype is poorly visible on the displayed images. Some improvements of the imaging parameters should be applied, for example by using phase contrast. Surprisingly, in another

panel of the same figure (Fig. 1E), fins of double mutants with the ET37-reporter seem to have a rather smooth margin, and no wrinkles can be observed, as far as I can see. Maybe the described abnormalities are dependent on genetic background?

5. What are Col2 aggregates in the double mutant?

Line 132 / Fig. S2B

Immunofluorescence analysis revealed a presence of abnormal Col2 aggregates in the double mutants. To prove the conclusion about the non-fibrillar organisation of this ECM protein, it is necessary to perform ultrastructure analysis by electron microscopy. This method, in any way, would be requested to determine if any actinotrichia-like collagenous striated bundles are present in the fin fold of larvae (see point 3).

6. What is the interplay between actinodin and collagen genes in actinotrichia formation?

Given that actinotrichia are composed of collagens and actinodins, it is possible that in the absence of Actinodin-1/2, the expression of the col2 gene is upregulated by a compensatory mechanism. Unfortunately, the study lacks any molecular analysis to understand what happens to the ECM gene expression in the double mutant.

7. What is the impact of Col2 aggregates on the hypural bone fusion?

Fig. 4 and 5.

The authors propose that actinotrichia have a direct role in guidance of hypural primordia. However, another explanation could be an indirect effect through Col2 aggregates, in the absence of Actinodin-1/2 proteins. Unfortunately, this phenotype remains unaddressed in the study. Given that Col2 is also expressed in the endoskeletal bones, it is possible that extensive accumulation of Col2 prevents the separation of individual hypurals. Thus, it remains unclear if actinotrichia are directly required for endoskeleton patterning, as proposed in the paper, or whether extensive Col2 aggregates bridge the adjacent hypurals, promoting their fusion? Accordingly, this aspect requires some closer examinations before drawing a conclusion about the role of actinotrichia in hypural formation.

8. What is the difference in cell elongation between WT and double mutant?

line 142-144, Fig 1E

The authors report about a difference in cellular elongation between wt and mutants, but the images display the tissue at an insufficient magnification to assess this change. Is the observed difference in length significant between both groups?

9. Where are actinotrichia in adult fin?

Figure 3

The issue is the same as in the case of the larval caudal fin: a visualization method of actinotrichia. Actinotrichia are arbitrarily drawn with red lines on images, but these data are not convincing. Actinotrichia must unambiguously be labelled using histological staining. Furthermore, histological and immunofluorescence stainings could be done on fin sections to better characterize the organization of the ECM in the mutants.

10. If adult mutants lack actinotrichia at the tip of the fin rays, what is replacing the missing fibers?

Figure 3

Col2 aggregates are at the tip of the ray. The authors conclude that Col2 fails to form fibers in the absence of And1/2. However, this conclusion is not convincing, as whole mount immunofluorescent staining lacks sufficient resolution. It would be necessary to perform ultrastructural analysis to investigate whether Collagen 2 forms fibers or remains disorganized in the mutant fin.

11. What is the link between actinotrichia, ray morphology and breeding tubercles in males?

Line 442-445, Figure 6

The finding that the double mutant males fail to stimulate egg release in females is interesting. In actinodin mutants, breeding tubercles (BTs) are irregular but, as far as I understand in Fig. 6, 25% of rays (11/44) had a normal pattern of these keratinized spikes. Furthermore, 40% of males (12/30) contained 3-4 rows of BTs, comparable to WT clusters. Are these males with normal BT clusters also inefficient in matings?

Overall, the statement that actinotrichia promote a pectoral fin ray morphology that is necessary for BT clusters is unclear. The effect is evidently indirect and maybe accumulation of Col2 or other matrix proteins simply disturb the normal patterning program. There is no convincing data showing that actinotrichia positively contribute to the patterning process, but maybe the ECM adjustment to the and1/2 deficiency triggers this phenotype. This hypothesis would explain a quite high ratio of normal BT clusters.

In conclusion, it would be important to develop a strategy to experimentally distinguish between the side-effects of Col2 aggregates in actinodin mutants versus the promoting function of actinotrichia in the normal fin tissue.

Minor comments

1. Normalization of fin morphology to standard length is poorly informative for comparison of fin size between WT and Mutants.

Fig. 1D. and Sup. Fig. S2D

I am not sure if the approach of data quantification as of % SL is optimal, given that there are no statistical variations of SL between groups, as shown in Sup. Fig. S2C.

The differences on graphs appear very small, such as approx. 0.02% for pectoral fin and 2% for caudal fin. It is difficult to interpret such a change. The authors could consider displaying measurements of the fin fold at each time point, without normalization to SL. In this way, the real difference in the fin size will be evident at all time points.

2. In double mutants, the difference of normalized fin length was up to 10% of the value in wild type, as shown in Fig. 2C. Thus, the claim that actinodin genes are essential for fin development, as written in the summary statement, should be rephrased.

3. Typo

Line 296: hadd

First revision

Author response to reviewers' comments

Response to reviewers

We would like to first thank the reviewers for their careful and thorough reading of our original manuscript. We made revisions and additions according to the insightful comments provided. We made several text modifications (detailed below) to improve clarity, and added new figures (for example, Fig S1, at the request of Reviewer 2) and new images to existing figures (for example, fluorescent immunohistochemistry) to better illustrate our claims. We have also adjusted our description of the significance of our work to not overinflate our findings. We included quantitative expression data at different time points to address potential ECM compensation mechanisms and histological staining on sections to gain a more detailed view of the tissues in question. We believe that these refinements and additions strengthen our manuscript and improve clarity for general audiences as well. We again thank the reviewers for the comments that led to these improvements. Below, we respond to the reviewers' comments specifically:

Reviewer 1

Specific comments:

1) *In Fig. 1E it is not easily to see what the authors mean. Elongation and arborization should be marked in a clear way.*

Our goal was to simply show that the mesenchymal cell migration is disorganized in the absence of actinotrichia. We used cell elongation and arborization merely as traits to describe this organization, and we did not intend to make direct claims about these traits. We reference Zhang

et al., 2010 to aid in describing ET37 WT fin fold mesenchymal cell migration, which has been previously described. To avoid confusion, we adjusted our language to more precisely reflect our intent:

164-168: “As mesenchymal cells migrate along the actinotrichia in the WT FFs, they take on an elongated shape, arranged in rows along the fibers (Zhang *et al.*, 2010) (Fig. 2C). In the double mutant FFs, the cells did not follow this linear migration pattern, and were arranged in a disorganized web, which did not occur in the WT FFs (Fig. 2C).”

We have also adjusted Fig. 2C (previously 1E) to include insets at a closer magnification to clearly show the migrating cells, and have added an arrow in the WT FFs to illustrate the direction of migration along the actinotrichia.

2) Lines 207-209: *Extension of rays past the distal extremity of the fin and reduced interray tissue are basically the same thing, no? Or is the idea that two independent processes are affected in double-mutants?*

We think that our wording “extension of rays past the distal extremity of the fin” was incorrect and might have introduced confusion. We apologize for that. We wanted to describe the effect of the absence of actinotrichia, on the one hand, on the morphology of the tip of the rays and on the other hand on the interray tissue. In WT, the bone matrix is tapering in the distal part of the ray. The actinotrichia are protruding from the bone matrix and provide a support for the distal edge of the fin tissue (as shown on the schematic in Figure S1 added to the revised version of the manuscript). In the double mutants, in absence of the support provided by the actinotrichia, the tip of the bone is no longer tapering and is directly underneath the distal epidermis (Fig 4). For the interray tissue, we observe a recession of the epidermis that is most likely a secondary effect of the absence of the actinotrichia in the rays. The fan-like organization of the actinotrichia array in the WT fins might help supporting and stretching the interray epidermis. Indeed, Nakagawa *et al.* (2022) reported that the actinotrichia provide a scaffold for the epidermal cells in the caudal fin at the tip of the rays, and provide structural support for dorsoventral or fan-like expansion. However, to avoid confusion, we reworded the explanation between lines 192-197:

192-197: “The double mutants’ rays were wider and wavy with irregularly spaced joints connecting each bone segment (Fig. 4B). At the tips of WT rays, a fan of actinotrichia supports the distal epidermal tissue of the fin (Fig. 4C,E). The double mutant rays end bluntly, rather than tapering distally to the extremity of the fin (Fig. 4D,F). The distalmost bone segments of double mutant rays are also cracked, suggesting higher susceptibility to damage (Fig. 4F). The interray tissue receded along the margin of the fin (Fig. 4D,F).”

Comments that need to be addressed in the discussion:

1) *The downregulation of and transcription in the mutants is curious. How do the authors interpret this finding? Is this an autoregulatory feature that can be overcome by injection of and mRNA? Are the remaining and genes not expressed in fins?*

We believe that the lack of strong staining with *and1* and *and2* probes by ISH in the mutants, despite having only partial deletions of the genes, is due to nonsense-mediated mRNA decay of aberrant transcripts encoding truncated proteins (Wittkopp *et al.*, 2009). We clarified this in the text:

139-141: “The single homozygous mutants, *and1^{-/-}and2^{+/+}* and *and1^{+/+}and2^{-/-}*, showed no expression in the FFs for the deleted gene (Fig. 1B), likely due to nonsense-mediated mRNA decay (Wittkopp *et al.*, 2009).”

358-359: “Partial deletions of *and1* and *and2* result in absent or severely reduced expression, likely due to nonsense-mediated mRNA decay (Wittkopp *et al.*, 2009).”

The *and3* and *and4* genes are also expressed in the fin folds of WT embryos (Zhang *et al.*, 2010). We have included RT-qPCR analysis of *and1-4* to evaluate their expression in 5 dpf WT siblings and double mutants (Figure 1C). *and1* and *and2* are significantly downregulated in the double mutants, consistent with our ISH data. There is no significant increase of *and3* in the double mutants, but we did note a significant upregulation of *and4*. However, the absolute expression of both *and3* and *and4* is quite low generally according to the ΔCq (Figure S3B), so we don’t think that the upregulation of *and4* is biologically significant. Later in development, at 14 dpf, there is no upregulation of *and3* and *and4* (Figure S8D). We think that *and1* and *and2* have a much more significant role than *and3* and *and4* for actinotrichia synthesis, given that perturbation of *and1* and *and2* is sufficient to prevent the synthesis of the fibers. We clarified this in the text as well:

147-151: “RT-qPCR analysis of relative *and1* and *and2* expression levels showed significant decreases in the double mutants relative to WT siblings at 5 dpf (Fig. 1C). There was no significant

upregulation in *and3*. Interestingly, *and4* was significantly upregulated (Fig. 1C), however the expression level is low for both WT and double mutant larvae (Fig. S3B).”

359-368: “The simultaneous loss-of-function of *and1* and *and2* resulted in the absence of actinotrichia from the FFs, even while shorter paralogous pair, *and3* and *and4*, were intact. These latter two are also expressed in the FFs during development (Zhang *et al.*, 2010), but encode much shorter proteins with fewer functionally important structural repeats (Zhang *et al.*, 2010). Interestingly, we did note a significant upregulation of *and4* at 5 dpf in double mutants relative to WT siblings (Fig. 1C). However, the normalized expression level of both *and3* and *and4* remains low in double mutants and WT siblings (Fig. S3B), so any compensatory upregulation is insufficient to rescue the defects caused by *and1* and *and2* disruption. Moreover, this upregulation seems temporary, as there is no upregulation of *and3* and *and4* later in development, at 14 dpf (Fig. S8D,E).”

2) *There is a triassic fossil, V. comptus, that has BT-like structures on its fins, suggesting that this type of courtship may have already been utilized by some ancient fish. However, this fossil is a member of the derived Neopterygian class and thus not a direct precursor of the lobe-finned fish from which the tetrapods derived. I would propose to tone down the claim that a shift in courtship strategy was required for the fin-to-limb transition (as highlighted in the summary statement).* We have adjusted our language from proposing that the loss of *actinodin* necessitated a shift in courtship strategy, to speculating that the loss of *actinodin* and a shift in courtship behaviour are related in the evolution of tetrapods, without knowing the order of these events.

Adjustments:

43-45: “We propose that *actinodin* gene maintenance is under strong selection in fish with similar courtship. We speculate that the loss of *actinodin* genes and a shift in courtship strategy may have coincided during tetrapod evolution.”

49-50: “Fish-specific *actinodin* genes are essential for proper fin patterning and reproductive success. *actinodin* gene loss in tetrapods relates to the fin-to-limb transition.”

447-454: “Although *V. comptus* occupies the more derived Neopterygian clade within Osteichthyes, it raises the possibility that a basal Osteichthyan fish living prior to the divergence of ray-finned fish and lobe-finned fish may have opted for a courtship mechanism involving BTs for external fertilization. Our findings suggest that fish using this strategy would experience a strong selective pressure to maintain their actinotrichia for proper patterning of their fin ray-associated BTs. We speculate that emergence of tetrapods may involve a shift in courtship behaviour, in addition to the loss of *actinodin* genes. However, the order of these events is unknown.”

464-466: “The loss of the *and* gene family during tetrapod evolution may have therefore occurred with a shift in courtship strategy independent of stimulation by pectoral breeding tubercles or other action from the fins.”

3) *If the idea was true that and genes are essential for courtship behavior, it might be interesting to examine the available fish genomes for the presence or absence of and genes. If there are examples where a loss has happened, it would be important to learn about their courtship rituals and fin development. Along these lines, it would be useful to speculate why either and1 or and2, although redundant in zebrafish fins, have not been eliminated by evolutionary processes.*

Bioinformatic analysis of *and* distribution in fish genomes is ongoing. All complete gnathostome genomes identified so far have *actinodin* orthologs, except tetrapods. It would indeed be valuable to examine any correlation between *actinodin* presence and BT-dependent courtship, and we have previously searched through the literature to address this. Detailed descriptions of fish courtship are generally scarce, but we did find a rare account of Australian lungfish courtship by Grigg (1965, Aust Nat Hist 30:50). The Australian lungfish, belonging to a sarcopterygian clade closely related to tetrapods, exhibits spawning behaviour that may not require the use of pectoral BTs. The male nudges the female’s cloaca with his snout, which is thought to stimulate oviposition. The breeding pair then release their gametes for external fertilisation in their preferred spawning site (Grigg, 1965). This account of lungfish spawning behaviour does not seem to require the physical stimulation of female egg release using pectoral fins. The presence of BTs on their fins is not specified, and these fish are not sexually dimorphic.

Our bioinformatic analyses (in progress, not included in the present manuscript) suggest no loss of *and* in gnathostomes other than in the tetrapod lineage. Zebrafish and other teleosts uniquely have four genes due to a teleost-specific whole genome duplication. As such, zebrafish have this redundancy due to this duplication. Why the two paralogs, *and1* and *and2*, are retained in the zebrafish genome is an interesting question. Although they show a strong and similar expression in

the developing fins, we previously showed that overstaining of *in situ* hybridization reveals unique expression for each gene in some distinct areas: for example, *and1* is uniquely expressed in the basal layer of the epidermis of the regenerating fin, *and2* is expressed in the embryonic heart and branchial arches (Zhang et al, 2010, supplemental information). Despite this, we did not observe obvious defects in the single mutants. However, it is possible that these unique expression patterns may explain the retention of the 2 copies on the teleost genome.

The idea the reviewer brings up interests us as well, but currently there is not sufficient information on courtship across diverse fish groups, or clear reports of sex and nuptial tubercle distribution for its inclusion in this manuscript. We do however look forward to examining this in our future projects.

4) Typo in line 296 (*hadd*).

Corrected.

Reviewer 2

The summary statement needs to be toned down to accurately reflect the reported findings.

We adjusted the summary statement to propose that *and* genes are important for proper fin patterning rather than general fin development. We have also adjusted statements about the significance throughout the text to more precisely reflect our intended claims. Please see response to similar comment by Reviewer 1.

The direct versus indirect phenotypes should be carefully defined to improve clarity and to avoid a risk of misinterpretation.

We have revised the text, differentiating between the direct effects of actinotrichia (such as those on FF and ray phenotype), indirect effects of actinotrichia (such as those on BT phenotype), and in our Discussion, expanded upon the role of actinotrichia on chondrogenesis and hypural formation (see response to comment 7 below).

The impact of the paper could be improved if some molecular mechanisms were included in the study, following the phenotypic characterization of the mutants.

We recognize that the impact of the absence of actinotrichia on embryonic fin development and bone patterning would need an analysis at the molecular level. This work is ongoing and will be the subject of another manuscript.

*The significance of the paper would potentially benefit by assessing not only fin development but also fin regeneration in the *and1/2* mutants.*

Work on fin regeneration of the double mutants is ongoing. We have not included it in the present manuscript because we have obtained unexpected and interesting results that need deeper analysis. We plan to report the fin regeneration data with the molecular analysis (previous comment) in a future manuscript.

Major specific points

1. *How do actinotrichia look like?*

Actinotrichia is the subject of this study.

Many readers of Development journal are not experts in skeletal biology of fish, being not yet familiar with these peculiar fibers. Thus, it would be helpful to start with an illustration to introduce the structure and distribution of actinotrichia.

We agree that a schematic of the organization of the actinotrichia in the embryos and in the adult fin ray would help non-expert readers. We added a new Figure S1 with schematics of actinotrichia in fin folds, and at the tips of the rays in adult fins.

2. *Where are actinotrichia in Figure 1?*

Line 127 / Fig. 1C

*The main conclusion of the paper is that the *and1/2* double mutants lack actinotrichia. However, the data for this statement are not convincing, because actinotrichia are not visualized in the specimens.*

Actinotrichia should be distinctively labeled to conclude about their presence or absence. On the image, 3 red lines seemingly depict actinotrichia. How have they been tracked on bright-field photographs of live zebrafish larvae? How many actinotrichia are expected in wild type embryos? If we cannot well see an actinotrichium in wild type, how can we see its absence in mutants?

To provide any evidence about these structures, it is necessary to perform histological staining for collagenous bundles of actinotrichia.

We have included clearer transmitted light images of actinotrichia in the FFs, with annotations that we hope will better orient the reader (Figure 2A, previously in Figure 1). We believe these images

more clearly show the radial arrangement of the actinotrichia populating the entire proximodistal distance of the WT and single mutant FFs at these stages, and the absence of this texture and organization in the double mutant FFs. We also included And1 and Col2 immunostaining of the FF. In Figure 2B, we show the fluorescent channel only to clearly show the signal from the antibodies, and since there was no signal for And1 in the double mutant, we included a transmitted light differential interference (TD)/fluorescent merge to depict that. We also added a complementary panel in Figure S3C with TD/fluorescent merge images for each antibody in WT and double mutants. We hope the immunostaining on FFs and makes the actinotrichia (and lack thereof in the double mutants) clear.

3. Do double mutants really lack any form of actinotrichia?

Beside actinodin 1 and 2, there is also actinodin 3 and 4, as mentioned in the introduction, which could contribute to actinotrichia.

Is it possible that the size and distribution of actinotrichia depend on the composition, specifically on the type of actinodins? If yes, shorter or thinner actinotrichia would be less conspicuous, and they could easily be overlooked in bright field or fluorescence images.

To support the main conclusion, it would be necessary to conduct ultrastructural analysis to prove the absence of any actinotrichia in the double mutant fin.

*and3 and and4 encode much smaller proteins than and1 and and2 with far fewer repeated motifs that are thought to play a role in actinotrichia formation (in zebrafish, And3 has 295 aa with 3 repeats, And4 has 306 aa with 4 repeats, vs And1 has 570 aa and 8 repeats, and And2 has 508 aa and 10 repeats). Available single-cell sequencing data on Danio cell shows that and1 and and2 are more strongly expressed during early development than and3 and and4 (Sur *et al.*, 2023, *Developmental Cell* 58(24): 3028-3047). We have now included RT-qPCR analysis of and1-4 (Figure 1C), which also show that and3 and and4 are not highly expressed. We did note a statistically significant upregulation of and4 in the double mutant at 5 dpf, however the absolute level of expression is still quite low (Figure S3B). We elaborated on this in lines 359-368 (see quote above). Our results show that the simultaneous loss-of-function of and1 and and2 produce a phenotype that is consistent with the hypothesized effect of the lack of actinotrichia: disorganized mesenchymal cell migration, FF fragility, smaller FF size, and defects in ray length and morphology, including the distal tip of each ray. These defects persist even though and3 and and4 are intact, suggesting that any contribution of these genes is not enough to compensate for the loss of and1 and and2. As such, we do not believe that the presence of and3 and and4 hinders our conclusions about the loss of actinotrichia due to and1 and and2 disruption. Of course, it would be interesting to finally describe the role of this smaller paralogous pair and contrast the relative importance of the four and genes in teleosts. This is one of our projects.*

Regarding the potential presence of smaller, thinner actinotrichia, we believe this confusion will benefit from a better definition of actinotrichia in the Introduction that we are now providing (lines 95-108). If there were smaller, thinner fibers present in the FF, it could be challenging to unambiguously categorize them as true actinotrichia. Actinotrichia also have known functions- to provide structural support to the developing fin and to serve as a scaffold for migrating mesenchymal osteoblast progenitors. The phenotype we describe in our double mutants shows disruption of these functions, consistent with the loss of actinotrichia.

Lines 95-108: “Elastoidin is distinguishable from collagen by its transparency, whereas collagen appears white (Ellis & McGavin, 1970), and tendency to form long needle-like fibers that behave as one single unit (McGavin & Pyper, 1964). Actinotrichia are huge fibers relative to the tissue they reside in; they can reach up to 100 μm in larval zebrafish FFs (Kuroda *et al.*, 2018), and increase in length during development to on average 260 μm at the tips of adult zebrafish caudal fin rays (Nakagawa *et al.*, 2022). Although the length and diameter vary depending on developmental stage (Durán *et al.*, 2011; Nakagawa *et al.*, 2022; Kuroda *et al.*, 2018), actinotrichia have a characteristic distribution in developing FFs and in the adult rays. Before ray formation, actinotrichia form two arrays of fibers, radially arranged approximately parallel to one another and span the entire proximodistal length of the FF (van den Boogart *et al.*, 2012). In adults, they overlap with the one or two distalmost bone segments of the rays, and extend past the bone, arranged in a distinct fan shape (Durán *et al.*, 2011; Pfefferli & Jazwinska, 2015). These distributions provide the overlaying epidermis with a texture that can be visualized with simple light microscopy (Fig. S1).”

Nonetheless, to better show the loss of actinotrichia from the tips of the rays, we have included Masson's Trichrome staining on 4-micron paraffin sections of the distal tips of rays (Figure S6). As

described in the response to the comment 2, actinotrichia were clearly stained in the WT rays, but no such structures were seen in the double mutants' rays. We opted for more detailed histology over ultrastructural analysis, as the latter is not routine in our lab and histological analysis is more easily accessible. We hope that the histological sections paired with better transmitted light images and immunostaining together provide a more convincing view of the presence and absence of actinotrichia in our WT and double mutants, respectively.

4. How severe and reproducible is the phenotype with the irregular margin of the fin and wrinkling of the tissue?

Line 129 Fig. 1C

The described phenotype is poorly visible on the displayed images. Some improvements of the imaging parameters should be applied, for example by using phase contrast. Surprisingly, in another panel of the same figure (Fig. 1E), fins of double mutants with the ET37-reporter seem to have a rather smooth margin, and no wrinkles can be observed, as far as I can see. Maybe the described abnormalities are dependent on genetic background?

We agree that (previous) Figure 1C did not clearly depict the actinotrichia. These images have been replaced with clearer ones (Figure 2A).

The comment about the phenotype discrepancy between 2A (previously 1C) and 2C (previously 1E) isn't clear to us. Both pffs are small and distally curled. Both mffs have a similar degree of visible truncation and neither is dramatically wrinkled. The wrinkling of the tissue in the ET37 double mutants' images is not as evident because of the nature of the image; the transmitted light image shows the texture of the FF, whereas the ET37 images are showing GFP signal in the mesenchymal cells within the FF. We hope that the phenotype is generally clearer with our improved images.

5. What are Col2 aggregates in the double mutant?

Line 132 / Fig. S2B

Immunofluorescence analysis revealed a presence of abnormal Col2 aggregates in the double mutants. To prove the conclusion about the non-fibrillar organisation of this ECM protein, it is necessary to perform ultrastructure analysis by electron microscopy. This method, in any way, would be requested to determine if any actinotrichia-like collagenous striated bundles are present in the fin fold of larvae (see point 3).

We understand that EM could reveal the presence/absence of collagenous striated bundles and be another line of evidence for the absence of actinotrichia. Unfortunately, EM is not a method we routinely use, We have experienced difficulties in the past with the EM service available to us, and as such we are reluctant to dedicate time and money to troubleshoot this approach. However, we have multiple lines of evidence using approaches we have much more experience with to illustrate the absence of actinotrichia in our mutants. They show that the functions that actinotrichia are involved in are all disrupted in the double mutant.

Calling the Col2 aggregates non-fibrous or non-fibrillar was poor wording on our part, as we do not know if the aggregate is fibrillar or not. We've adjusted the language, for example in the Figure 5 caption, to remove inferences about the fibrous or non-fibrous nature of the aggregates.

6. What is the interplay between actinodin and collagen genes in actinotrichia formation?

Given that actinotrichia are composed of collagens and actinodins, it is possible that in the absence of Actinodin-1/2, the expression of the col2 gene is upregulated by a compensatory mechanism. Unfortunately, the study lacks any molecular analysis to understand what happens to the ECM gene expression in the double mutant

We included RT-qPCR analysis of *and1-4*, *col2a1a* and *col2a1b* using 5 dpf and 14 dpf larval tails to address the potential of compensation for the disruption of *and1* and *and2*. *col2a1a* and *col2a1b* were analyzed because they encode the proteins making up Type II Collagen, another major component of actinotrichia, in zebrafish. *col2a1b* has been shown to be expressed in the embryonic FFs (Durán *et al.*, 2011). Although there is no clear report of *col2a1a* expression in the FF, we included it in our analysis as it is a duplicated paralog of *col2a1b* and encodes a constituent of Type II Collagen. We did not report any significant differences in *col2a1a* or *col2a1b* expression levels between WT siblings and double mutants (Figure 1C, Figure S8D). Because *col2a1a* is expressed in the cranial cartilages and the notochord, we took care to use tissue that would highlight fin fold expression levels and minimize detection of expression in other tissues- we cut the larvae posterior to the urogenital pore. We also noted a nonsignificant upregulation of *and3* and a significant

upregulation of *and4* at 5 dpf (Figure 1C), however the expression level is still rather low (Figure S3C), and evidently does not compensate for the phenotypes we observed in the double mutants. This upregulation also does not persist throughout development, as we did not detect significant differences in *and3*, *and4*, *col2a1a*, or *col2a1b* expression at 14 dpf (Fig. S8D). As such, it does not appear that the Type II Collagen-encoding genes are compensating for the decrease in *and1* and *and2*. We added more information about Col2a1's roles in cartilage development and expanded on the potential crosstalk between actinotrichia and cartilage development through collagen:

402-413: “Indeed, extracellular scaffolds, like actinotrichia, can provide anchoring sites for chondrocytes (Aigner & Stove, 2003). Col2a1 itself can act as a signalling molecule in chondrocyte metabolism (Garamzegi *et al.*, 2010; Xin *et al.*, 2015), acting through chondrocytes' integrin receptors for Col2a1 (Enomoto *et al.*, 1993). Actinotrichia, containing fibrillar Col2a1, could therefore have a direct role in chondrocyte behaviour in the developing hypurals. However, actinotrichia may also simply provide structural support and spatial restriction for chondrogenesis. Although actinotrichia and the hypural cartilages overlap spatially in WT larvae (Fig. 5I,J), in the absence of actinotrichia, Col2 proteins aggregate within the FF, away from the hypurals developing on the ventral side of the notochord, with no spatial overlap (Fig. S8C). There was also no significant upregulation of the genes encoding zebrafish Type II collagen, *col2a1a* and *col2a1b* prior to (Fig. 1C) or during hypural development (Fig. S8D).”

7. What is the impact of Col2 aggregates on the hypural bone fusion?

Fig. 4 and 5.

The authors propose that actinotrichia have a direct role in guidance of hypural primordia. However, another explanation could be an indirect effect through Col2 aggregates, in the absence of Actinodin-1/2 proteins. Unfortunately, this phenotype remains unaddressed in the study. Given that Col2 is also expressed in the endoskeletal bones, it is possible that extensive accumulation of Col2 prevents the separation of individual hypurals. Thus, it remains unclear if actinotrichia are directly required for endoskeleton patterning, as proposed in the paper, or whether extensive Col2 aggregates bridge the adjacent hypurals, promoting their fusion? Accordingly, this aspect requires some closer examinations before drawing a conclusion about the role of actinotrichia in hypural formation.

In our Col2 immunostaining analyses, we never observed an overlap in the Col2 aggregations and the condensing mesenchyme giving rise to the hypural cartilages. The aggregations did not appear to collect around the hypurals, but instead are located farther (as seen in Figure 4 and Figure S8C). The Col2 signal is also not consistently brighter in the double mutants' hypurals, suggesting that they do not take up more Col2, which could be the case in the scenario suggested. We have clarified this by including a panel containing additional Col2-immunostained WT and double mutant larvae, to show the consistency of this phenotype (Fig. S8C). The Col2 aggregations have no spatial overlap with the hypurals in the double mutants. This shows that the Col2 is not bridging the hypurals and contributing to their fusions. We believe that the fusions are an indicator of disorganized hypural growth.

We added clarification in the Discussion to clearly offer both possibilities suggested by the reviewer: the loss of actinotrichia could lead to the hypural fusions through disorganization during development, or through impaired chondrocyte organization (lines 402-413, passage already included above).

8. What is the difference in cell elongation between WT and double mutant?

line 142-144, Fig 1E

The authors report about a difference in cellular elongation between wt and mutants, but the images display the tissue at an insufficient magnification to assess this change.

Is the observed difference in length significant between both groups?

We adjusted Figure 2C (previously 1E) to include insets at a higher magnification to more clearly show the mesenchymal cells. We used cell elongation as a trait to describe the organization and shape of the cells, but did not intend to imply any quantitative difference, as we did not measure the cells. We believe the images are more impactful than statistics in this case, since we are showing that in the presence of actinotrichia, the mesenchymal cells clearly follow a straight, regular migration along the actinotrichia, whereas without them, the cells lack a clear angle of migration.

We also adjusted our language in the associated Results section to precisely reflect our intent (lines 164-168, passage quoted above).

9. Where are actinotrichia in adult fin?

Figure 3

The issue is the same as in the case of the larval caudal fin: a visualization method of actinotrichia. Actinotrichia are arbitrary drawn with red lines on images, but these data are not convincing.

Actinotrichia must unambiguously be labelled using histological staining.

Furthermore, histological and immunofluorescence stainings could be done on fin sections to better characterize the organization of the ECM in the mutants.

We have replaced certain panels in Figure 3 with clearer transmitted light images of the ray tip in WT siblings and double mutants. We hope that the actinotrichia in the WT are more evident in these images (Figure 4E), and that this makes their absence easier to see in the double mutant (Figure 4F). We also added images of And1- and Col2-immunostaining, showing that the signal of these antibodies localizes to the fibers, thus labelling the actinotrichia in the WT (Figure 4G-J, Figure S5C). We hope this labelling is unambiguous to readers as well.

We included paraffin sections at three different levels along the proximal-distal axis of WT and double mutant rays stained with Masson's Trichrome that stains structures containing collagen in blue (Figure S6). The selected positions are representative of a distalmost level of a WT fin that does not contain bone matrix and only actinotrichia (Figure S6B,C), a more proximal level showing two rows of actinotrichia located between the two hemirays (Figure S6D,E), and at an even more proximal level, only hemirays are visible (Figure S6F,G). Sections of the double mutant rays at equivalent positions only show the blue staining corresponding to the bone matrix of the hemirays, which is thicker than in WT fins, but does not reveal the presence of actinotrichia. Schematic representations of the selected sections are also shown in the schematic in Figure S1.

10. If adult mutants lack actinotrichia at the tip of the fin rays, what is replacing the missing fibers?

Figure 3

Col2 aggregates are at the tip of the ray. The authors conclude that Col2 fails to form fibers in the absence of And1/2. However, this conclusion is not convincing, as whole mount immunofluorescent staining lacks sufficient resolution. It would be necessary to perform ultrastructural analysis to investigate whether Collagen 2 forms fibers or remains disorganized in the mutant fin.

We would like to clarify that we believe the Col2 signal at the tips of the double mutant rays is not an aggregate, but rather synthesized Col2 that accumulates abnormally when not taken up into actinotrichia. We apologize for the poor wording. We opted to perform histological staining of transverse paraffin sections of the rays to gain a more detailed view inside the ray tips (Figure S6). Interestingly, we did not see any accumulation of collagenous material in the distalmost sections of the double mutants' rays (which would have been stained blue by Masson's Trichrome stain), despite seeing the Col2 signal in the whole mount IHC. We noted that this Col2 signal was fainter than the Col2 aggregation in the larval FFs. Masson's trichrome stains collagen, however it does not specifically recognize any particular type, therefore we cannot tell if the thicker structure of the hemirays is uniquely the Type I Collagen found in bone, or also Type II Collagen. The histological staining does show the absence of organized actinotrichia fibers from the double mutant rays, and no additional disorganized ECM was detected so far. Even if Col2 forms fibers, it is not forming actinotrichia as we have defined them based on the current literature (see lines 95-108). We believe these additional experiments are sufficient in showing the presence/absence of fibers, along with the existing analyses. EM is costly and these above methods are readily available to us.

11. What is the link between actinotrichia, ray morphology and breeding tubercles in males?

Line 442-445, Figure 6

The finding that the double mutant males fail to stimulate egg release in females is interesting. In actinodin mutants, breeding tubercles (BTs) are irregular but, as far as I understand in Fig. 6, 25% of rays (11/44) had a normal pattern of these keratinized spikes. Furthermore, 40% of males (12/30) contained 3-4 rows of BTs, comparable to WT clusters. Are these males with normal BT clusters also inefficient in matings?

Yes, males that have some normal BT clusters are also inefficient in mating. No double mutant male has been successful in reproducing using the characteristic zebrafish mating behaviour; 6 double mutant males were formally tested, as shown in Fig. S9A (but the real number is much higher in the years we have been investigating these mutants). We would like to clarify that the indicated sample size concerns the BT cluster on a particular ray, so even though some clusters may be normal on double mutants, not all clusters within the fin had this phenotype. Even if some rays had normal clusters, most rays in double mutant pectoral fins were still abnormal. The number of clusters in the double mutant males being similar to WT males supports that the BTs develop

normally, but that they have defects at the patterning level, which is related to the ray phenotype that is directly affected by the absence of actinotrichia. However, this comment highlights the fact that we cannot tease apart if the inability to breed is more to do with the BTs being abnormally patterned, or rather to do with the general pectoral fin defects (shorter fin, fewer rays, malformed rays). For example, some double mutant males are still able to execute their hooking behaviour and grasp the female, suggesting that there may be another factor needed to stimulate egg release. We already remarked the former possibility in our original version (see lines 431-433), but we made adjustments to our language in the Discussion to accommodate the latter possibility more:

Line 420-421: “Double mutant males are unable to stimulate egg release in females during courtship, which we generally attribute to their pectoral fin ray defects.”

In our original manuscript version, we already included the following sentence in our Discussion to offer the two possibilities that could explain the double mutants’ inability to breed:

Lines 426-429: “Although it is possible that the reduced fin size and ray defects prevent males from stimulating female egg release, the double mutant males also have abnormal pectoral fin BTs, which have been posited to be involved in reproduction in other fish as well (Wiley & Collette, 1970; Kortet *et al.*, 2004; Kortet *et al.*, 2003; Yamamoto & Egami, 1974).”

Overall, the statement that actinotrichia promote a pectoral fin ray morphology that is necessary for BT clusters is unclear. The effect is evidently indirect and maybe accumulation of Col2 or other matrix proteins simply disturb the normal patterning program. There is no convincing data showing that actinotrichia positively contribute to the patterning process, but maybe the ECM adjustment to the and1/2 deficiency triggers this phenotype. This hypothesis would explain a quite high ratio of normal BT clusters.

We agree that the BT phenotype in the double mutants is an indirect effect of the lack of actinotrichia through their direct effect on ray patterning. We interpreted the presence of normal BT clusters in the double mutants as variation in phenotype severity due to the indirect relationship of actinotrichia and BT patterning. The indirect nature of this relationship is also illustrated by non-fin BTs appearing normal in the double mutant males.

In conclusion, it would be important to develop a strategy to experimentally distinguish between the side-effects of Col2 aggregates in actinodin mutants versus the promoting function of actinotrichia in the normal fin tissue.

It could be interesting to generate a triple mutant line with deletions in *and1*, *and2*, and *col2a1b*, the zebrafish *col2a1* paralog that is thought to contribute specifically to actinotrichia (Durán *et al.*, 2011). Perturbing the actinodin and collagen components of actinotrichia synthesis could give insight into the effects of the loss of actinotrichia compared to the effects of Col2 aggregation. However, disrupting collagen is challenging as collagens have important functions beyond actinotrichia, and their perturbation may introduce new phenotypes that are not directly related to actinotrichia function. A conditional KO of *col2a1b* in the cells of the fins would also be challenging in zebrafish. The advantage of perturbing actinodin function is that the encoding genes seem to be specific in function to actinotrichia without any observable off-target effects.

We would like to clarify that the Col2 aggregation is a direct effect of the absence of actinotrichia. As such, we believe there is no separate effect of Col2 aggregates. The hypural fusions do not seem to be “side effects” of the aggregates, as the Col2 aggregates do not overlap spatially with the hypural cartilages (Fig. S8C), but rather accumulate away from the hypurals.

Minor comments

1. Normalization of fin morphology to standard length is poorly informative for comparison of fin size between WT and Mutants.

Fig. 1D. and Sup. Fig. S2D

I am not sure if the approach of data quantification as of % SL is optimal, given that there are no statistical variations of SL between groups, as shown in Sup. Fig. S2C.

The differences on graphs appear very small, such as approx. 0.02% for pectoral fin and 2% for caudal fin. It is difficult to interpret such a change. The authors could consider displaying measurements of the fin fold at each time point, without normalization to SL. In this way, the real difference in the fin size will be evident at all time points.

We normalized the fin size to SL to control for the variations in SL throughout development. Although the differences were not significant, we believe this normalization more accurately and robustly represents the size of the fin at each stage when making comparisons. Nonetheless, we have included un-normalized measurements in addition to the normalized measurements, and the data show similar results as after normalization to SL (now in Figure S4).

2. In double mutants, the difference of normalized fin length was up to 10% of the value in wild type, as shown in Fig. 2C. Thus, the claim that *actinodin* genes are essential for fin development, as written in the summary statement, should be rephrased.

The difference in fin length was statistically significant, demonstrating that the absence of *actinotrachia* does have an effect on fin outgrowth- the mutant phenotype was evidently abnormal and this was supported by statistical analysis. We would like to clarify that we do not intend to claim that *and* genes are essential for fin development, but rather they are essential for *proper* fin development that is then crucial for reproductive success in zebrafish. In the summary statement (line 49), we previously said that “Fish-specific *actinodin* genes are essential for fin development and reproductive success.” This was an oversight that we have corrected to the following: “Fish-specific *actinodin* genes are essential for proper fin patterning and reproductive success.” We revised the rest of the text to ensure we say only that the *and* genes contribute to fin development, that *actinotrachia* influence development, or that they are essential for *proper* fin development.

3. Typo Line 296: *had*

Corrected.

Modifications made to accommodate new changes and manuscript word count (line numbers refer to highlighted manuscript document):

Lines 400-409: Although the observation of the smaller hypural diastema in 6/8 double mutants is interesting, we believe it is more parsimoniously a consequence of smaller fin size and crowding during development rather than a phenotype of evolutionary significance. We do however add a passage about the connective tissue plates related to the diastema in lines 225-227.

Lines 426-430: These lines were redundant with lines 244-248.

Figure 1 in the original submission was divided into Figures 1, 2, and S4.

Figure 2 in the original submission became Figure 3 in the revised manuscript. Original Figure 3 became revised Figure 4, with some imaging improvements.

Figure 4 in the original submission was moved to the Supplemental Material as Supplemental Figure S7.

Second decision letter

MS ID#: dev.204990R1

MS TITLE: An Essential Role for *Actinotrachia* in Zebrafish Fin Patterning and Courtship Behaviour

AUTHORS: Paulina Hanzelova, Connor Baird, Bidemi Keshinro, Reeham Kadhom, Robert Lalonde and Marie-Andree Akimenko

Dear Marie-Andree,

I have now received both referees' reports on the above manuscript, and have reached a decision. The referees' comments are appended below.

As you will see, while one referee recommends acceptance of the manuscript, the other considers that your revisions have not sufficiently addressed concerns raised and recommends further revisions prior to consideration for publication. If you are able to revise the manuscript along the lines suggested, I will be happy to receive a further revised version of the manuscript. If it would

be helpful, you are welcome to contact us to discuss your revision in greater detail. Please send us a point-by-point response indicating your plans for addressing the referees' comments, and we will look over this and provide further guidance.

Please attend to all of the reviewer's comments and ensure that you upload both a 'clean' version of your Word file, along with a highlighted version clearly showing where you have made changes in the revised manuscript. Please avoid using 'Tracked changes' in Word files as these are lost in PDF conversion. I should be grateful if you would also provide a point-by-point response detailing how you have dealt with the points raised by the reviewers in the 'Response to Reviewers' box. If you do not agree with any of their criticisms or suggestions please explain clearly why this is so.

Reviewer 1

Advance summary and potential significance to field

This manuscript addresses interesting questions for the community interested in limb development, the role of actinodins in fin development beyond the embryo. The authors show that actinodin genes are important to guide molecular processes in adult teleost fin development, and discover that actinotrichia in the fin fold guide the formation of cartilaginous precursors of the hypural bones of the caudal fin endoskeleton. They propose an evolutionary scenario in which the loss of actinodin genes in tetrapods was accompanied by a reduction of the fin dermoskeleton.

Comments for the author

The manuscript looks fine.

Reviewer 2

Advance summary and potential significance to field

This manuscript makes several novel contributions to our understanding of fin patterning through characterization of a new double mutant line lacking actinotrichia. The authors discovered that actinotrichia loss impairs male reproductive success through defective breeding tubercle (BT) patterning, an unexpected reproductive phenotype. They demonstrate that actinotrichia influence not just dermal rays but also endochondral bone development, resulting in hypural fusions. This reveals a new developmental role for actinotrichia. The authors propose a scenario linking actinodin gene loss to changes in courtship strategy and present intriguing evolutionary hypotheses.

Comments for the author

The revised study shows improvement and the manuscript is stronger than the previous version. However, some critical points have not yet been adequately addressed and require careful revisions.

1. Several major mechanistic gaps remain unaddressed in this study, preventing this work from advancing our understanding of the findings in a meaningful way. The two key findings are reported without mechanistic insights, undermining the impact of the paper:
 - A. The connection between actinotrichia loss and hypural fusions is largely correlative. The authors suggest that actinotrichia provide "structural support" or "spatial restriction" for chondrogenesis, but direct evidence is lacking. Do actinotrichia actively signal to chondrocytes, or is this purely a physical function? If yes, then by what means? What is the nature of spatial restriction or structural support? This point is critical and needs mechanistic explanation.
 - B. The link between ray defects and BT patterning abnormalities is assumed but not demonstrated. Are BTs responding to altered ray morphology, reduced mechanical properties, or disrupted signaling? The correlation remains unexplained.
2. Evolutionary interpretations must be toned down as they sound speculative, or the authors should provide more comparative data.

A. The link between actinodin gene loss and courtship strategy shift in tetrapod evolution is highly speculative. The authors acknowledge uncertainty about temporal order but do not discuss alternative scenarios or how this hypothesis could be tested.

B. Discussion of fossils is interesting, but broader comparison across extant teleosts with different courtship strategies would strengthen the reproductive selection argument. Findings are most directly applicable to cypriniform fishes. Generalizability to other teleost groups needs examination.

C. The evolutionary hypotheses on the fin-to-limb transition and morphology-behavior link, while intriguing, remain largely untestable with current data. The concept that the zebrafish fin represents an ancestral origin of the tetrapod limb should be avoided, as teleosts represent a specialized branch of vertebrate evolution.

Minor comment: Sample sizes are not always clearly stated in figure legends for all measurements.

Second revision

Author response to reviewers' comments

Reviewer 1: SUMMARY OF THE ADVANCE MADE IN THIS PAPER AND ITS POTENTIAL SIGNIFICANCE TO THE FIELD

This manuscript addresses interesting questions for the community interested in limb development, the role of actinodins in fin development beyond the embryo. The authors show that actinodin genes are important to guide molecular processes in adult teleost fin development, and discover that actinotrichia in the fin fold guide the formation of cartilaginous precursors of the hypural bones of the caudal fin endoskeleton. They propose an evolutionary scenario in which the loss of actinodin genes in tetrapods was accompanied by a reduction of the fin dermoskeleton.

SUGGESTIONS TO AUTHORS

The manuscript looks fine.

We thank Reviewer 1 for their appreciation of our revised manuscript. We are pleased that our updates sufficiently addressed the comments.

Reviewer 2: SUMMARY OF THE ADVANCE MADE IN THIS PAPER AND ITS POTENTIAL SIGNIFICANCE TO THE FIELD

This manuscript makes several novel contributions to our understanding of fin patterning through characterization of a new double mutant line lacking actinotrichia. The authors discovered that actinotrichia loss impairs male reproductive success through defective breeding tubercle (BT) patterning, an unexpected reproductive phenotype. They demonstrate that actinotrichia influence not just dermal rays but also endochondral bone development, resulting in hypural fusions. This reveals a new developmental role for actinotrichia. The authors propose a scenario linking actinodin gene loss to changes in courtship strategy and present intriguing evolutionary hypotheses.

SUGGESTIONS TO AUTHORS

The revised study shows improvement and the manuscript is stronger than the previous version. However, some critical points have not yet been adequately addressed and require careful revisions.

We thank Reviewer 2 for their careful reading of our updated manuscript and corresponding responses. We appreciate the effort and advice to improve our work and maximise its impact. Please see our responses to the specific comments below.

1. Several major mechanistic gaps remain unaddressed in this study, preventing this work from advancing our understanding of the findings in a meaningful way. The two key findings are reported without mechanistic insights, undermining the impact of the paper:

A. The connection between actinotrichia loss and hypural fusions is largely correlative. The authors suggest that actinotrichia provide "structural support" or "spatial restriction" for chondrogenesis, but direct evidence is lacking. Do actinotrichia actively signal to chondrocytes, or is this purely a physical function? If yes, then by what means? What is the nature of spatial restriction or structural support? This point is critical and needs mechanistic explanation.

In the present manuscript, we wanted to describe the effect of perturbation of *actinodin* on fin development. We show that the deletions in *and1* and *and2* lead to the lack of actinotrichia synthesis, which has negative effects on the development of the skeleton forming within the fin fold, which includes the hypurals. We then show that the lack of actinotrichia does not seem to impair the initiation of chondrogenesis in the hypurals, suggesting that they do not influence the actual process of chondrogenesis, but rather impact the subsequent organization and proper growth of the hypurals. Altogether, this suggests that it is the organization of the chondrocytes that is affected. This is a similar case as with the other cell types located between the two actinotrichial arrays in the embryonic fin folds; their migration and organization is perturbed by the absence of actinotrichia. Based on the literature, actinotrichia provide a physical scaffold for the developing fin rays (Wood, 1982; Wood & Thorogood, 1984; Dane & Tucker, 1985; Nakamura *et al.*, 2016; Lee *et al.*, 2013). There is no evidence that actinotrichia signal to the surrounding cells, however they have been shown to physically provide guidance for cell migration (Wood & Thorogood, 1984; Kuroda *et al.*, 2018, Kuroda *et al.*, 2020; König *et al.*, 2018). We discussed the nature of this relationship already in the Discussion (lines 391-396). Regarding the hypurals developing among the actinotrichia in the fin fold, it seems that they, like the rays, are affected by the lack of actinotrichia, resulting in disorganization and malformation.

B. The link between ray defects and BT patterning abnormalities is assumed but not demonstrated. Are BTs responding to altered ray morphology, reduced mechanical properties, or disrupted signaling? The correlation remains unexplained.

We interpreted our results to mean that the clusters of BTs have defects most likely due to the defects in ray morphology, rather than disrupted signalling. The BTs form at puberty onset (Dai *et al.*, 2021), on already-formed rays. In the double mutants, where these rays had defects, the BT clusters were mispatterned; although they occasionally appeared normal on some rays analyzed (11/44 rays with BTs), most rays had abnormal BT clusters (the remaining 33/44 rays with BTs). However, BTs within the clusters appeared normal based on histological analysis. On other fins where BTs are found (although not in clusters), the BTs also had patterning abnormalities. BTs found on non-fin tissue (and thereby not influenced by actinotrichia during development) appear normal in the double mutants. These results show that double mutants have BT patterning abnormalities that are fin-specific. Since proper fin morphology is influenced by actinotrichia, we argue that the double mutants' BT patterning defects are an indirect effect of the lack of actinotrichia on ray morphology. We already described these results in more detail in lines 291-308. We believe this explanation is most parsimonious based on the lines of evidence we provided. A main idea of this manuscript is that the absence of actinotrichia leads to defects of the fin rays, including the BT clusters forming on the males' pectoral fin rays, which in turn explains our unexpected observation of the males' inability to breed. We demonstrate this idea by characterized the ray defects (Figures 3, 4, S5, S6) and the BT cluster defects (Figure 6) in double mutants, and show their effects on courtship behaviour with video evidence (Video 1). The mechanism by which the BT clusters form abnormally in the double mutants is beyond the scope of the present manuscript.

2. Evolutionary interpretations must be toned down as they sound speculative, or the authors should provide more comparative data.

A. The link between *actinodin* gene loss and courtship strategy shift in tetrapod evolution is highly speculative. The authors acknowledge uncertainty about temporal order but do not discuss alternative scenarios or how this hypothesis could be tested.

We clearly state that we are speculating this point (lines 44-45, 440-441, 453-455). This

idea could be strengthened with descriptions of courtship of fish belonging to other clades (and we have previously searched the literature), however information on their behaviour and BT distribution is scarce. We have also made revisions based on similar comments by Reviewer 1 (comments 2 and 3) in the previous revision, which they have deemed sufficient.

B. Discussion of fossils is interesting, but broader comparison across extant teleosts with different courtship strategies would strengthen the reproductive selection argument. Findings are most directly applicable to cypriniform fishes. Generalizability to other teleost groups needs examination.

We would like to clarify that we are not attempting to generalize a trend to all fish. We did specify that the actinotrichia are important for reproductive success in “zebrafish and other cypriniforms using a pectoral fin and/or BT-mediated courtship” (lines 451-453), which we believe specifies our result to the most relevant subject, rather than extrapolating to teleosts.

C. The evolutionary hypotheses on the fin-to-limb transition and morphology-behavior link, while intriguing, remain largely untestable with current data. The concept that the zebrafish fin represents an ancestral origin of the tetrapod limb should be avoided, as teleosts represent a specialized branch of vertebrate evolution.

We agree that the zebrafish fin is relatively derived and does not represent the fin morphology that gave rise to tetrapod limbs. However, zebrafish are much more practical and feasible as a model organism than fish that would be more representative, such as gars or sarcopterygian fishes. The derived character of the zebrafish fin is a caveat for evolutionary developmental biology studies using zebrafish as a model (such as Hawkins *et al.*, 2021; Kherdjemil *et al.*, 2016; Cadete *et al.*, 2023; Nakamura *et al.*, 2016). However, it is important to explicitly state this so that it is clear for readers of diverse backgrounds. We have added a passage in our discussion to address this limitation of our study (lines 442-445):

“Zebrafish also have a relatively derived fin morphology (Coates, 1994; Coates & Cohn, 1998; Grandel & Schulte-Merker, 1998; Lalonde & Akimenko, 2018), and occupy a derived phylogenetic position relative to the fish ancestral to tetrapods. As such, zebrafish studies will benefit from complementary comparative analyses.”

Minor comment: Sample sizes are not always clearly stated in figure legends for all measurements.

We added the missing sample sizes to the Figure 5 legend to reflect the numbers noted in the panels (lines 976-977).

Works cited in response

- Cadete, F., Francisco, M., & Freitas, R. (2023). Bmp-signaling and the finfold size in zebrafish: implications for the fin-to-limb transition. *Evolution*, 77(5), 1262-1271.
- Dai, X., Pu, D., Wang, L., Cheng, X., Liu, X., Yin, Z., & Wang, Z. (2021). Emergence of breeding tubercles and puberty onset in male zebrafish: evidence for a dependence on body growth. *Journal of Fish Biology*, 99(3), 1071-1078.
- Dane, P. J., & Tucker, J. B. (1985). Modulation of epidermal cell shaping and extracellular matrix during caudal fin morphogenesis in the zebra fish *Brachydanio rerio*. *Development*, 87(1), 145-161.
- Hawkins, M. B., Henke, K., & Harris, M. P. (2021). Latent developmental potential to form limb-like skeletal structures in zebrafish. *Cell*, 184(4), 899-911.
- Kherdjemil, Y., Lalonde, R. L., Sheth, R., Dumouchel, A., de Martino, G., Pineault, K. M., ... & Kmita, M. (2016). Evolution of Hoxa11 regulation in vertebrates is linked to the pentadactyl state. *Nature*, 539(7627), 89-92.

- König, D., Page, L., Chassot, B., & Jaźwińska, A. (2018). Dynamics of actinotrichia regeneration in the adult zebrafish fin. *Developmental biology*, 433(2), 416-432.
- Kuroda, J., Itabashi, T., Iwane, A. H., Aramaki, T., & Kondo, S. (2020). The physical role of mesenchymal cells driven by the actin cytoskeleton is essential for the orientation of collagen fibrils in zebrafish fins. *Frontiers in Cell and Developmental Biology*, 8, 580520.
- Kuroda, J., Iwane, A. H., & Kondo, S. (2018). Roles of basal keratinocytes in actinotrichia formation. *Mechanisms of Development*, 153, 54-63.
- Lee, R. T. H., Knapik, E. W., Thiery, J. P., & Carney, T. J. (2013). An exclusively mesodermal origin of fin mesenchyme demonstrates that zebrafish trunk neural crest does not generate ectomesenchyme. *Development*, 140(14), 2923-2932.
- Nakamura, T., Gehrke, A. R., Lemberg, J., Szymaszek, J., & Shubin, N. H. (2016). Digits and fin rays share common developmental histories. *Nature*, 537(7619), 225-228.
- Nakamura, T., Gehrke, A. R., Lemberg, J., Szymaszek, J., & Shubin, N. H. (2016). Digits and fin rays share common developmental histories. *Nature*, 537(7619), 225-228.
- Wood, A. (1982). Early pectoral fin development and morphogenesis of the apical ectodermal ridge in the killifish, *Aphyosemion scheeli*. *The Anatomical Record*, 204(4), 349-356.
- Wood, A., & Thorogood, P. (1984). An analysis of in vivo cell migration during teleost fin morphogenesis. *Journal of cell science*, 66(1), 205-222.

Third decision letter

MS ID#: dev.204990R2

MS TITLE: An Essential Role for Actinotrichia in Zebrafish Fin Patterning and Courtship Behaviour

AUTHORS: Paulina Hanzelova, Connor Baird, Bidemi Keshinro, Reeham Kadhom, Robert Lalonde and Marie-Andree Akimenko

Dear Marie-Andree,

I sent your manuscript back to reviewer 2 who is now happy with your revisions and so I am happy to tell you that your manuscript has been accepted for publication in *Development*, pending our standard publication integrity checks.

Reviewer 2

Advance summary and potential significance to field

Comments for the author

I would like to thank the authors for their thorough and careful revision of the manuscript. They have addressed all of the concerns and questions raised in the previous review in a clear and satisfactory manner.

I have no further concerns regarding this manuscript. In its current form, I believe it meets the standards of the journal and makes a meaningful contribution to the field. I therefore recommend acceptance of the manuscript for publication without further revisions.

I congratulate the authors on their work and look forward to seeing this study published.